# Negative Spatial Autocorrelation: One of the Most Neglected Concepts in Spatial Statistics

## Daniel A. Griffith 

School of Economic, Political, and Policy Sciences, University of Texas at Dallas, 800 W. Campbell Road, Richardson, TX 75080, USA; dagriffith@utdallas.edu; Tel.: +1-972-883-4950

**Abstract:** Negative spatial autocorrelation is one of the most neglected concepts in quantitative geography, regional science, and spatial statistics/econometrics in general. This paper focuses on and contributes to the literature in terms of the following three reasons why this neglect exists: Existing spatial autocorrelation quantification, the popular form of georeferenced variables studied, and the presence of both hidden negative spatial autocorrelation, and mixtures of positive and negative spatial autocorrelation in georeferenced variables. This paper also presents details and insights by furnishing concrete empirical examples of negative spatial autocorrelation. These examples include: Multi-locational chain store market areas, the shrinking city of Detroit, Dallas-Fort Worth journey-to-work flows, and county crime data. This paper concludes by enumerating a number of future research topics that would help increase the literature profile of negative spatial autocorrelation.

**Keywords:** hidden spatial autocorrelation; Moran coefficient; positive-negative spatial autocorrelation mixture; spatial competition

## 1. Introduction

Quantitative data analysis researchers routinely initiate their studies by examining univariate features of their data, such as frequency distribution skewness, the relationship between a mean and its accompanying variance, and the presence of outliers (e.g., see reference [1]). If these researchers study two or more variables, they also habitually inspect pairwise linear correlation coefficients. The topic of this paper falls under the heading of this second data analysis practice, with particular emphasis on negative correlation.

As most introductory statistics textbooks demonstrate, a linear correlation may be classified as being positive, zero (i.e., some narrow interval of values centering on and approximately equal to zero), or negative. Let X and Y be two variables, whose observed values for a sample of size n may be denoted respectively by $x_i$ and $y_i$, i = 1, 2, ... , n. Very explicitly, positive linear correlation means that, relatively speaking, high values of X, $x_H$, tend to pair with high values of Y, $y_H$, medium values of X, $x_M$, tend to pair with medium values of Y, $y_M$, and low values of X, $x_L$, tend to pair with low values of Y, $y_L$. Zero linear correlation means that $x_H$ tends to pair with $y_H$, $y_M$, and $y_L$, $x_M$ tends to pair with $y_H$, $y_M$, and $y_L$, and $x_L$ tends to pair with $y_H$, $y_M$, and $y_L$; no high, medium, or low value pairing preferences emerge. Negative linear correlation means that $x_H$ tends to pair with $y_L$, $x_M$ tends to pair with $y_M$, and $x_L$ tends to pair with $y_H$. This description, well-known to statisticians, is noteworthy in order to use it as a foundation for the ensuing discussion. Figure 1a furnishes a cross-tabulation for n = 254 illustrating a negative correlation (it is roughly −0.5 in this example), the topic of this paper. Its color coding enables the establishment of a parallel for describing correlation associated with Figure 1b, which relates to spatial statistics, a more recent subdiscipline addition to the statistics literature.

| Y \ X | L | M | H |
|-------|------|------|------|
| H | **43** | 32 | 9 |
| M | 24 | **36** | 26 |
| L | 17 | 18 | **49** |

X: the 2010 percentage of rural population [= 100x (rural population count)/(total population count)] in a Texas county

Y: the 2010 population density [= (total population count)/area] of a Texas county

(**a**)

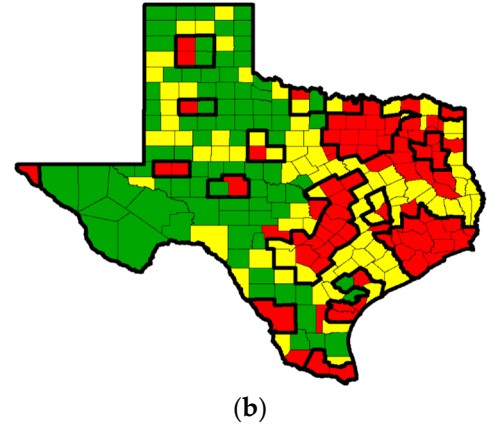

(**b**)

**Figure 1.** Correlation illustrations; variable tretile value groupings [red denotes high (H), yellow denotes medium (M), and green denotes low (L) values]. Left (**a**): A cross-tabulation of paired values bivariate frequencies. Right (**b**): A choropleth map of 2010 Texas population density by county (demarcated by thin black borders), with thick black borders outlining census metropolitan regions.

Many other forms of correlation exist in addition to the traditional linear bivariate Pearson product moment one describing associations, mostly between pairs of variables. Some are specific to particular measurement scales (e.g., nominal and ordinal); others are specific to studied themes. These latter include autocorrelation, which literally means self-correlation, and refers to correlation among observation values of a single variable, a fundamental topic of spatial statistics. Perhaps the most common autocorrelation investigated is for time series, and it is serial in structure. Another type, namely spatial autocorrelation (see reference [2]), is geographic in structure. The measurement value pairings of traditional linear correlation are by observation: A unit of analysis possesses two variable attributes. The measurement value pairings of spatial autocorrelation pertain to data organization and are juxtaposed locations: Two neighboring areal units possess correlated values for a single attribute. One difference here is that linear correlation is bivariate, whereas spatial autocorrelation is univariate. Paralleling the preceding description of linear correlation, for some variable Y, positive spatial autocorrelation (PSA) means that, relatively speaking, $y_H$ tends to neighbor $y_H$, $y_M$ tends to neighbor $y_M$, and $y_L$ tends to neighbor $y_L$: Neighboring values are relatively similar (see Figure 1b; green counties tend to be neighbors, yellow counties tend to be neighbors, and red counties tend to be neighbors). Zero spatial autocorrelation means that $y_H$ tends to neighbor $y_H$, $y_M$, and $y_L$, $y_M$ tends to neighbor $y_H$, $y_M$, and $y_L$, and $y_L$ tends to neighbor $y_H$, $y_M$, and $y_L$. Negative spatial autocorrelation (NSA) means that $y_H$ tends to neighbor $y_L$, $y_M$ tends to neighbor $y_M$, and $y_L$ tends to neighbor $y_H$: Neighboring values are relatively dissimilar. A perusal of relevant textbooks and internet web pages reveals the presentation of very few real world NSA examples. This concept is one of the most—if not the most—neglected spatial statistics topics, and as such is the motivation for and theme of this article. Griffith [3] and Griffith and Arbia [4] furnish the initial conceptual and foundational discussions about NSA; this paper differs from their work by introducing new traits of NSA (e.g., its relation to jackknifing and model misspecification), beginning with an emphasis on special features of negative correlation in general. Therefore, the primary purpose of this paper is to furnish evidence supporting why statisticians should care about this mostly ignored concept.

Figure 1a portrays moderate negative bivariate correlation. Meanwhile, Figure 1b portrays the most commonly presented spatial autocorrelation case, namely moderate positive; in particular, the Dallas, Houston, and Austin-San Antonio metropolitan areas are conspicuous on this Texas map. A checkerboard is the most commonly presented NSA case; it is sufficiently illustrative but not truly anchored in real world data.

## 1.1. Visualizing Correlation

Quantitative researchers exploring univariate characteristics of data frequently employ histograms, box-plots, and quantile plots, among other statistical graphics. Extending analysis beyond a single variable, they regularly employ scatterplots to explore bivariate relationships. A scatterplot is a two-dimensional graph portraying plotted paired values of two variables, X and Y, with reference to its axes (which may be expressed as z-scores); the pattern of the resulting cloud of points reveals the nature and degree of any correlation present (see Figure 2). The key data organizing notion here is the assignment of X and Y values to each i. If the trend line summarizing the scatter of points is linear, then its slope relates to the correlation coefficient describing the nature and degree of the relationship between X and Y. The linear Pearson product moment correlation coefficient may be written as

$$r = \frac{\sum_{i=1}^{n}(y_i - \bar{y})(x_i - \bar{x})/(n-1)}{\sqrt{\sum_{i=1}^{n}(y_i - \bar{y})^2/(n-1)}\sqrt{\sum_{i=1}^{n}(x_i - \bar{x})^2/(n-1)}} = \sum_{i=1}^{n} z_{y_i} z_{x_i}/(n-1),$$

where $z_{p_i}$ denotes a z-score for observation i and variable p. The plotted points may be the z-score pairs $(z_{y_i}, z_{x_i})$; converting the $(x_i, y_i)$ pairs of points to their z-score equivalents converts the slope of the scatterplot trend line to r. Situations like Figure 2c are the focus of this paper.

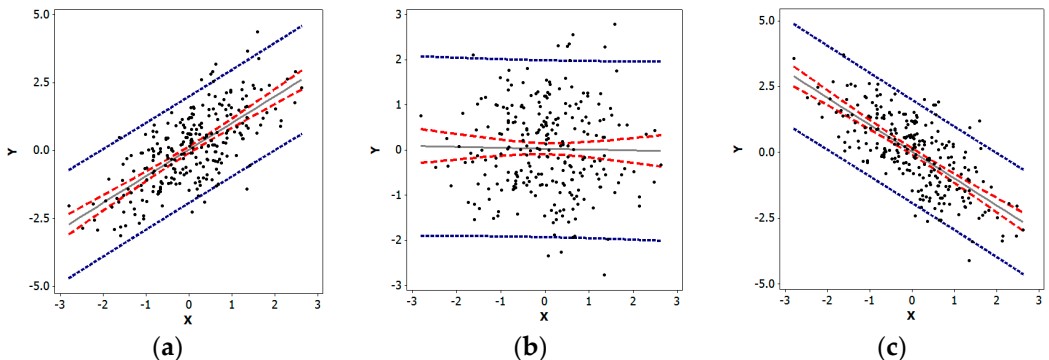

**Figure 2.** Specimen scatterplots portraying different natures and degrees of bivariate linear correlation (n = 254) using z-score axes; blue denotes 95% prediction intervals, red denotes 95% confidence intervals, and gray denotes trend lines. Left (**a**): Positive, r ≈ 0.7. Middle (**b**): Zero, r ≈ 0. Right (**c**): r ≈ −0.7.

Various indices exist that measure spatial autocorrelation (e.g., the Moran Coefficient (MC), the Geary Ratio, and join counts). The one most similar to Pearson's r is the MC; both involve covariation (i.e., their numerator terms). Similar to r, MC can be rewritten to allow the construction of a scatter diagram known as a Moran scatterplot, which is a two-dimensional graph in which each value of a single variable Y and the sum of its neighboring values are plotted along its axes, with the pattern of the resulting cloud of points revealing any nature and degree of spatial autocorrelation present (see Figure 3). The key data organizing notion is the n-by-n spatial weights matrix (e.g., see reference [5]), **C**, that indicates which locations are neighbors; this matrix articulates the data's organization. In its simplest form, the entries of this matrix are $c_{ij} = 1$ if the matrix row and column label areal unit polygons share a common non-zero length boundary, and $c_{ij} = 0$ otherwise; sometimes analysts convert this matrix to its row-standardized version, **W** [i.e., $w_{ij} = c_{ij}/\sum_{j=1}^{n} c_{ij}$]. Making an analogy with chess, this neighbors designation is known as the rook definition of geographic adjacency; a number of other definitions exist (e.g., the queen, making another analogy with chess, which also includes zero length boundaries (i.e., points), and the nearest neighbors, which may be the set of polygons whose centroids are closest to the centroid of a given polygon). If the trend line representing the scatter of points is linear, then its slope relates to the MC describing the nature and degree of the spatial autocorrelation contained in variable Y. This spatial autocorrelation coefficient may be written as

$$\text{MC} = \frac{n}{\sum_{i=1}^{n}\sum_{j=1}^{n}c_{ij}}\frac{\sum_{i=1}^{n}\sum_{j=1}^{n}c_{ij}\left(y_i - \overline{y}\right)\left(y_j - \overline{y}\right)}{\sum_{i=1}^{n}\left(y_i - \overline{y}\right)^2} = \frac{n}{(n-1)\sum_{i=1}^{n}\sum_{j=1}^{n}c_{ij}}\sum_{i=1}^{n}z_i\sum_{j=1}^{n}c_{ij}z_j. \quad (1)$$

Comparing Equation (1) with the formula for r reveals that it replaces X with Y, and it averages its numerator covariation terms over the number of neighbors $\sum_{i=1}^{n}\sum_{j=1}^{n}c_{ij}$ rather than n. One noteworthy feature of this index is that its extreme values are not necessarily ± 1; instead, they are determined by the extreme eigenvalues of a specifically modified version of matrix **C** (see §1.3). Now the plotted points may be the z-score pairs $(z_i, \sum_{j=1}^{n}c_{ij}z_j)$. In this context, the slope of the trend line relates to the MC. Situations like the one exemplified by Figure 3c are the focus of this paper. Furthermore, a goal of this paper is to demonstrate that NSA exists in disparate empirical datasets, at least more commonly than believed, and needs to be better understood; its study is more than merely an academic curiosity.

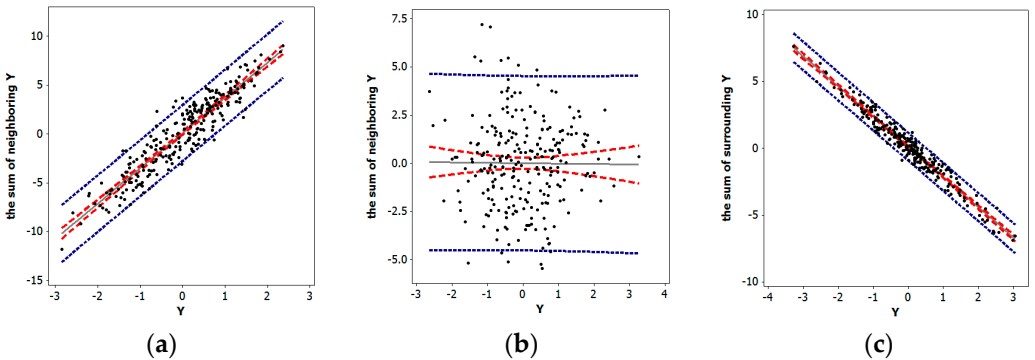

**Figure 3.** Specimen Moran scatterplots portraying different natures and degrees of spatial autocorrelation (n = 254) using z-score axes; blue denotes 95% prediction intervals, red denotes 95% confidence intervals, and gray denotes trend lines; $\text{MC}_{max}$ = 1.098 and $\text{MC}_{min}$ = −0.635 denote the extreme Moran coefficient (MC) values. Left (**a**): Positive, $\text{MC}/\text{MC}_{max} \approx 0.7$. Middle (**b**): Zero, MC ≈ 0. Right (**c**): $\text{MC}/|\text{MC}_{min}| \approx −0.7$.

*1.2. What Is Special About Negative Correlation?*

The literature is replete with examples of negatively correlated variables; the preceding relationship between Texas percentage rural population and population density (Figure 1a) is one. An immediate consequence of this abundance of occurrences is a lack of attention devoted to nuances, idiosyncrasies, and distinctive properties attributable to the nature of negative correlation, with no indication of any effort to compile a collection of these properties. Nevertheless, the following five special facets are noteworthy:

1.  Linear regression residuals are negatively correlated;
2.  Multinomial indicator variables yield a correlation matrix with all negative off-diagonal entries;
3.  Positively skewed independent variables have a strong tendency to display negative bivariate correlation;
4.  Negatively correlated replicates can reduce variance in simulation experiments; and,
5.  Negative bivariate correlation can be relative.

The objective of discussing these five items in this section is to make their collective existence better acknowledged, establishing underpinnings for extending the notion of special to NSA.

A well-established property is that linear regression residuals are negatively correlated. Consider the simple case of the intercept-only regression model specification, which may be written as follows:

$$\textbf{Y} = a\textbf{1} + \textbf{e,}$$

where **Y** is an n-by-1 vector of variable values, **1** is an n-by-1 vector of ones, a = $\bar{y}$ is the ordinary least squares estimate of the regression coefficient, and **e** is an n-by-1 vector of residuals (**Y** − a**1**). The estimate $\bar{y}$ places a linear constraint on the residuals (i.e., $\sum_{i=1}^{n-1} e_i = -e_n$) that results in the expected value $E(e_i e_j) = -1/(n-1)$, where $E(\bullet)$ denotes the calculus of expectations, even with the standard assumption of independent regression errors.

For the next trait, consider a set of mutually exclusive and collectively exhaustive binary 0–1 indicator variables, such as those associated with one-way analysis of variance (ANOVA). The correlation matrix for this set of variables has ones in its diagonal, and off-diagonal (h, k) row-column (i.e., h ≠ k) entries of $-\sqrt{n_h n_k / [(n-n_h)(n-n_k)]}$, where $n_j$ denotes the number of one's indicator variable j contains. For the case of two groups (i.e., a binomial random variable), this off-diagonal formula always reduces to −1. For the 16 census metropolitan regions of Texas (see Figure 1b; the 17th indicator variable would be for the non-metropolitan counties), the correlation matrix is

| | | | | | | | | | | | | | | | | |
|---|---|---|---|---|---|---|---|---|---|---|---|---|---|---|---|---|
| 1.00 | −0.10 | −0.19 | −0.14 | −0.14 | −0.14 | −0.17 | −0.17 | −0.10 | −0.19 | −0.17 | −0.10 | −0.14 | −0.24 | −0.35 | −0.35 | −0.41 |
| −0.10 | 1.00 | −0.01 | −0.01 | −0.01 | −0.01 | −0.01 | −0.01 | 0.00 | −0.01 | −0.01 | 0.00 | −0.01 | −0.01 | −0.01 | −0.01 | −0.02 |
| −0.19 | −0.01 | 1.00 | −0.01 | −0.01 | −0.01 | −0.01 | −0.01 | −0.01 | −0.02 | −0.01 | −0.01 | −0.01 | −0.02 | −0.03 | −0.03 | −0.03 |
| −0.14 | −0.01 | −0.01 | 1.00 | −0.01 | −0.01 | −0.01 | −0.01 | −0.01 | −0.01 | −0.01 | −0.01 | −0.01 | −0.01 | −0.02 | −0.02 | −0.02 |
| −0.14 | −0.01 | −0.01 | −0.01 | 1.00 | −0.01 | −0.01 | −0.01 | −0.01 | −0.01 | −0.01 | −0.01 | −0.01 | −0.01 | −0.02 | −0.02 | −0.02 |
| −0.14 | −0.01 | −0.01 | −0.01 | −0.01 | 1.00 | −0.01 | −0.01 | −0.01 | −0.01 | −0.01 | −0.01 | −0.01 | −0.01 | −0.02 | −0.02 | −0.02 |
| −0.17 | −0.01 | −0.01 | −0.01 | −0.01 | −0.01 | 1.00 | −0.01 | −0.01 | −0.01 | −0.01 | −0.01 | −0.01 | −0.02 | −0.03 | −0.03 | −0.03 |
| −0.17 | −0.01 | −0.01 | −0.01 | −0.01 | −0.01 | −0.01 | 1.00 | −0.01 | −0.01 | −0.01 | −0.01 | −0.01 | −0.02 | −0.03 | −0.03 | −0.03 |
| −0.10 | 0.00 | −0.01 | −0.01 | −0.01 | −0.01 | −0.01 | −0.01 | 1.00 | −0.01 | −0.01 | −0.00 | −0.01 | −0.01 | −0.01 | −0.01 | −0.02 |
| −0.19 | −0.01 | −0.02 | −0.01 | −0.01 | −0.01 | −0.01 | −0.01 | −0.01 | 1.00 | −0.01 | −0.01 | −0.01 | −0.02 | −0.03 | −0.03 | −0.03 |
| −0.17 | −0.01 | −0.01 | −0.01 | −0.01 | −0.01 | −0.01 | −0.01 | −0.01 | −0.01 | 1.00 | −0.01 | −0.01 | −0.02 | −0.03 | −0.03 | −0.03 |
| −0.10 | 0.00 | −0.01 | −0.01 | −0.01 | −0.01 | −0.01 | −0.01 | −0.00 | −0.01 | −0.01 | 1.00 | −0.01 | −0.01 | −0.01 | −0.01 | −0.02 |
| −0.14 | −0.01 | −0.01 | −0.01 | −0.01 | −0.01 | −0.01 | −0.01 | −0.01 | −0.01 | −0.01 | −0.01 | 1.00 | −0.01 | −0.02 | −0.02 | −0.02 |
| −0.24 | −0.01 | −0.02 | −0.01 | −0.01 | −0.01 | −0.02 | −0.02 | −0.01 | −0.02 | −0.02 | −0.01 | −0.01 | 1.00 | −0.04 | −0.04 | −0.04 |
| −0.35 | −0.01 | −0.03 | −0.02 | −0.02 | −0.02 | −0.03 | −0.03 | −0.01 | −0.03 | −0.03 | −0.01 | −0.02 | −0.04 | 1.00 | −0.05 | −0.06 |
| −0.35 | −0.01 | −0.03 | −0.02 | −0.02 | −0.02 | −0.03 | −0.03 | −0.01 | −0.03 | −0.03 | −0.01 | −0.02 | −0.04 | −0.05 | 1.00 | −0.06 |
| −0.41 | −0.02 | −0.03 | −0.02 | −0.02 | −0.02 | −0.03 | −0.03 | −0.02 | −0.03 | −0.03 | −0.02 | −0.02 | −0.04 | −0.06 | −0.06 | 1.00 |

For this example, $n_j$ ranges from 1 to 17; the rounded correlation coefficients range from −0.004 ≈ 0 to −0.41. Usually constructing a correlation matrix with all negative off−diagonal entries is a challenging feat; encountering such matrices is rare in practice.

Beardsle [6] reports a fascinating discovery that positively skewed independent variables have a strong tendency to display negative correlation; Ladson [7] summarizes a simulation experiment (n = 100; r = 1000; Pearson Type III variables) supporting this contention. The simulation experiment summarized here (see Table 1) confirms and extends Ladson's findings. Its entries involve selected sample sizes of 100 to 1000 for batches of 1000 r values, with 1000 replications each. The percentages of r < 0 range from roughly 52% to 75%, consistent with Ladson's reported results. This experiment included a varying sample size for the best (i.e., exponential random variable) and worst (i.e., Weibull random variable) cases, to investigate whether or not this property is a small sample phenomenon. These preliminary results suggest an ultimate asymptotic convergence on 50%, but at a very slow rate. One check performed during this simulation experiment monitored the value of $\bar{r}$, which was always within the confidence interval of 0, sometimes greater than it, and sometimes less than it. In contrast, the percentages of batches with a majority of r < 0 cases were usually 100%, which was not within their binomial-based confidence intervals. The selected Weibull random variable has a skewness that exceeds 60, much higher than the largest skewness used by Ladson. As a benchmark, Beardsle comments that hydrological variables often have skewness in excess of seven.

Exploiting a serendipitous advantage of negative correlation dates back to Hammersley and Morton's [8] proposal to create antithetic variates when designing a simulation experiment, although this simulation experimental design element is not without criticism (e.g., reference [9]). This technique involves creating a complementary replication for each random replication in a Monte Carlo experiment such that each random selection $y_i$ in a given replication has a corresponding $-y_i$ in its complement

replication. The feature of interest is a comprehensive covering of the employed probability space: $y_i$ covers part of it, whereas $-y_i$ covers its complement, a different part of the probability space. In other words, the first sequence of random numbers is reused in a simulation experiment. The mathematical statistics property of interest pertains to sums/differences of random variables:

$$\text{VAR}(Y_1 + Y_2) = \text{VAR}(Y_1) + \text{VAR}(Y_2) + 2\text{COV}(Y_1, Y_2)$$
$$= \text{VAR}(Y_1) + \text{VAR}(Y_1) - 2\sqrt{\text{VAR}(Y_1)}\sqrt{\text{VAR}(Y_1)} \text{ here,}$$

where VAR denotes the variance of a random variable, and COV denotes the covariance of two random variables; the correlation between the random draw of $Y_1$ and its antithetic variable $Y_2$ is a perfect $-1$, by construction. The benefit of using this antithetic variable replication is variance reduction, as shown by the preceding mathematical statistics theorem (i.e., fewer replications are needed to achieve a smaller standard error of, for example, an arithmetic mean); this variance reduction is between random draws, not the simulated statistic of interest. Another advantage is that statistics of interest are based upon more regularly distributed sample values, allowing better balancing of more extreme simulated pseudo-random numbers.

**Table 1.** Summary results for a simulation experiment examining the frequency of r <0 for independent and identically distributed variables.

| Random Variable | Parameters | Skewness | n | % of r < 0 Batches | Average % of r < 0 |
|---|---|---|---|---|---|
| exponential | $\lambda = 1$ | 2.00 | 100 | 92 | 52.2 |
| | | | 500 | 77 | 51.2 |
| | | | 1000 | 70 | 50.9 |
| beta | $\alpha = 0.3, \beta = 25$ | 3.39 | 100 | 100 | 56.0 |
| gamma | $\alpha = 0.1, \beta = 1$ | 4.47 | 100 | 100 | 59.0 |
| log-normal | $\mu = 0, \sigma^2 = 1$ | 6.18 | 100 | 100 | 56.0 |
| | | | 100 | 100 | 75.0 |
| Weibull | $\lambda = 0.25, \kappa = 1$ | 60.09 | 500 | 100 | 73.0 |
| | | | 1000 | 100 | 72.0 |

Another peculiarity of negative correlation is that it can be relative. This situation arises in both principal components and factor analysis. Normalized eigenvectors of correlation matrices, used to construct principal components and factor scores, are unique except for a multiplicative factor of $-1$. Because the loadings/structure matrix tabulates the correlations between original variables and these synthetic scores, signs in these matrices flip when eigenvectors are multiplied by $-1$. In other words, although two variables with different loading signs for a given component/dimension are negatively correlated, the natures of their correlations with that component/dimension are relative.

In summary, negative bivariate correlation exhibits at least five prominent special advantageous insights not furnished by positive correlation alone. Negatively correlated linear regression residuals in practice accompany the assumption of independent residuals in theory. The 0–1 indicator variables affiliated with all multinomial random variables are pairwise negatively correlated. Positively skewed independent variables tend to yield an overwhelming preponderance of negative r values. Negatively correlated Monte Carlo simulation replications purportedly improve a simulation experiment's performance. Finally, sometimes the negative sign of r is relative.

*1.3. What Is Special About NSA?*

NSA is a fundamental spatial statistical concept, although only a relatively few academic treatments thoroughly address it, perhaps because its relative frequency of empirical occurrences is widely believed to be low. For example, it is found in only eight of 361 (i.e., 2.2%) agricultural plant breeding field trials [10], and 80 of 2801 (i.e., 2.9%) United States (US) intra-county population density geographic distributions [11]. Unfortunately, few other empirical NSA examples appear in the literature; Chun and

Griffith [12] furnish a short but nearly exhaustive list of them to date. Besag [13], for example, also rebuffed NSA by proving that auto-Poisson and auto-negative binomial (NB) random variables, among others, can accommodate only it, whereas most geographic distributions of counts display PSA. Consequently, NSA currently is one of the most neglected topics in spatial statistics. However, similar to negative bivariate correlation, it exhibits special qualities, including

1.  NSA manifestations differ between discrete and continuous geographic space;
2.  NSA links to spatial competition;
3.  Common spatial autocorrelation indices tend to gauge NSA on a scale shorter than [−1,0];
4.  Extreme NSA supports the fast calculation of the extreme eigenvalues of certain matrices;
5.  The boundary between PSA and NSA for the MC is zero rather than some small negative value; and,
6.  NSA often mixes with PSA, which tends to mask its existence.

This section addresses these six particulars, with the goal of fostering a greater appreciation of NSA and an awareness of its substantive existence.

One discerning difference between negative bivariate correlation and spatial autocorrelation is their relationships to their positive counterparts. Observations are always discrete for bivariate correlation, whereas spatial autocorrelation can involve discrete (polygons) or ambiguously demarcated continuous (points separated by distance) observations. When in the continuous domain, NSA cannot occur in adjacent (i.e., separated by an infinitesimally small distance) locations; rather, it can materialize only at a sizeable distance. The trigonometric sine-based wave-hole semi-variogram trend line (Figure 4a) is one of the few geostatistical models [14] that describes this type of circumstance. This descriptor highlights that in the continuous domain, local (i.e., fine geographic scale and resolution) spatial autocorrelation must be positive. The possibility of local NSA defies logic because two values negatively correlated at a distance d cannot also be negatively correlated with a single intervening value at, say, d/2. Meanwhile, as Figure 4a implies, if polygon boundaries are set at appropriate distances, NSA could be masked (e.g., it is averaged with PSA in its geographic aggregate, within an areal unit polygon), or NSA could materialize (e.g., a polygon boundary encircles only the initial PSA). Accordingly, the focus of this paper is on polygonal areal units created by surface partitioning's, such as administrative districts (e.g., counties, states), census geography (e.g., census tracts, census blocks), and remotely sensed satellite images (e.g., raster pixels).

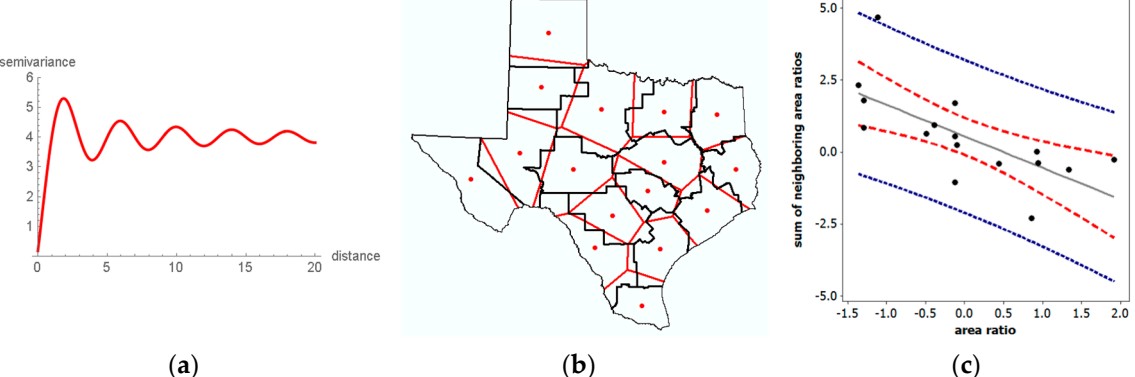

**Figure 4.** Examples involving NSA. Left (**a**): A wave-hole semi-variogram model plot. Middle (**b**): Texas environmental quality service regions with their superimposed Thiessen polygon counterparts. Right (**c**): A z-score axes Moran scatterplot for the polygon-to-region area ratio of Figure 4b; MC = −0.274.

NSA naturally relates to spatial competition processes. One regularly cited example describes a row of plants growing in a garden. Because of genetics, for example, these plants begin growing at different times and grow at different rates, some get larger faster, consequently hoarding water

and sunlight (and perhaps even nutrients) to the detriment of their neighboring plants. The resulting harvest amounts tend to be characterized by NSA. Mead [15] noted this condition for the weights of neighboring carrot and cabbage plants as the distance separating them decreased. The macaw parrot furnishes a similar illustration. A female macaw lays two or three eggs during an interval of several-days, in part to ensure the survival of offspring. If the baby hatching first survives, then it is overpoweringly strong when its siblings hatch and it hoards the food delivered by its parents (very local spatial competition) to the extent that its siblings almost always die. This scenario coupled with its outcome was an initial justification for establishing Peru's Tambopata Research Center. River piracy provides yet another real world example. In 2016, the retreat of the Kaskawulsh Glacier in the Yukon resulted in all of the water constituting the Slims River being diverted to the nearby Alsek River, causing the Slims River to vanish while increasing the Alsek River's flow by a factor of 60 to 70. One final noteworthy illustration concerns the crime displacement hypothesis. When law enforcement increases in a jurisdiction, crime in that district tends to decrease, whereas crime in surrounding jurisdictions tends to increase. Furthermore, the examination of, say, crime by police precinct within this jurisdiction most likely would uncover PSA (i.e., all crime tends to decrease, although perhaps at differential rates). Figure 4b embodies the NSA represented by these four examples. Consider the empirical Texas regions, defined by the black line borders. The red dots are the geometric centroids of these regions. Thiessen polygons (i.e., a polygon containing every surface point closer to its red centroid location than to any of the other 15 red locations—Thiessen polygons are created from a finite set of discrete focal points by connecting nearby focal point pairs with lines, and then constructing the polygon boundaries with perpendicular bisectors of these connecting lines; each created polygon defines a catchment area around its focal point such that any location inside this polygon is closer to that point than to any of the other focal points; the resulting surface partitioning is planar) overlay Texas to identify the most compact set of rival polygons to partition the state's surface. The quantity of interest here is the ratio of the two polygon areas (area of a Thiessen polygon)/(area of its associated region). A Thiessen polygon can gain area vis-à-vis its corresponding region only by annexing parts of adjacent regions; a Thiessen polygon can lose area only by forfeiting parts of its corresponding region's periphery to adjacent regions. The Moran scatterplot in Figure 4c confirms that this surface repartitioning is a spatially competitive process. This example signifies that one informative use of NSA can be as an index of regionalization scheme compactness. This contention is bolstered by Figure 5a, a similar Thiessen polygon exercise with the 16 counties forming the northwest part of Texas. Because these counties are nearly perfectly compact, and the superimposed Thiessen polygons almost exactly coincide with 12 of them, the slight reallocation of the area produces near-zero positive, rather than more substantial negative, spatial autocorrelation.

A value of −0.274 for the previous geographic area reallocation example (Figure 4c) seems to imply a weak negative relationship; in other words, weak NSA. However, as emphasized in the preceding discussion, the extreme values of the MC are not necessarily ± 1, but rather functions of the extreme eigenvalues of the following modified matrix **C** expression, which is the matrix version of the non-attribute variable Y portion of the numerator of Equation (1):

$$(\mathbf{I} - \mathbf{11}^{\mathrm{T}}/n)\mathbf{C}(\mathbf{I} - \mathbf{11}^{\mathrm{T}}/n), \tag{2}$$

where **I** denotes the n-by-n identity matrix. Pre- and post-multiplication of the spatial weights matrix **C** by the projection matrix $(\mathbf{I} - \mathbf{11}^{\mathrm{T}}/n)$ constitutes the aforementioned specific modification. This matrix operation converts the principal eigenvalue of matrix **C** to zero, and its corresponding principal eigenvector (this pair of mathematical quantities, one a scalar and the other a vector, is an eigenfunction) to a vector proportional to **1**; the remaining n − 1 eigenfunctions are approximately unchanged, although all are transformed to have a mean of zero (i.e., each is orthogonal to the vector **1**). When multiplied by $n/\mathbf{1}^{\mathrm{T}}\mathbf{C1}$, the n eigenvalues of matrix expression (2) become MC values, indexing the nature and degree of spatial autocorrelation portrayed by the mapping of their corresponding eigenvectors. For the Texas surface partitioning, the minimum MC is −0.610; Figure 5b portrays

its corresponding eigenvector, which depicts the maximum NSA map pattern possible for any set of real numbers combined with the Texas counties spatial weights matrix **C** based upon the rook adjacency definition. This minimum value indicates that −0.274 effectively is equivalent to −0.449 (≈ −0.274/|−0.610|), a quantity calculated by stretching the endpoint of the MC scale to the more intuitive and appealing limit of −1; in other words, the NSA is moderate, not weak. Regular square tessellation surface partitionings, such as Figure 5a, relate to yardsticks whose extremes are closer to ± 1; for this 4-by-4 tessellation, the minimum possible MC is −1.079 (see Figure 5c). The minimum for most irregular surface partitioning is closer to −0.5. Furthermore, frequency distributions of eigenvalues, and hence MC values reveal that far more potential NSA than PSA map patterns exist (see Figure 5c). The theoretical NSA upper limit is 75% of the n eigenvalues, which requires spatial weights matrices built using planar graphs comprising only n = 4 completely connected (known as $K_4$) subgraphs. The practical upper limit is roughly 67%. The apparent empirical upper limit for a reasonable size n (e.g., ≥100) is approximately 60%. All three of these scenarios contain far more distinct NSA than PSA map patterns.

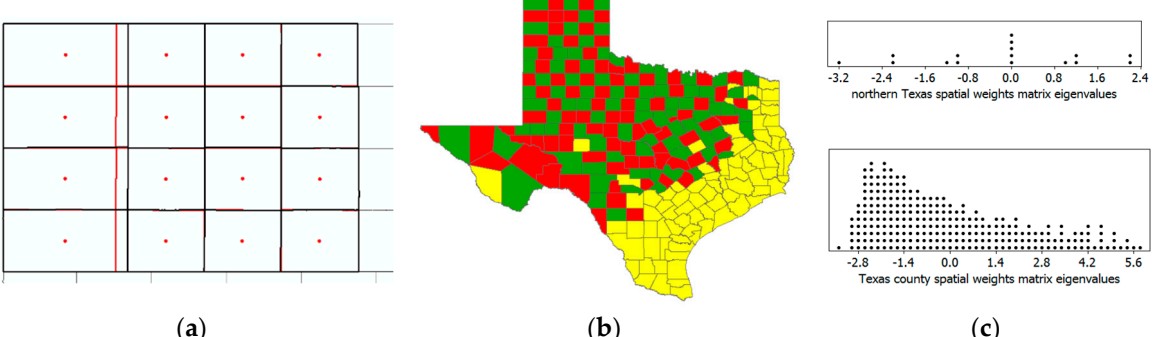

(**a**)                                   (**b**)                                   (**c**)

**Figure 5.** NSA benchmark exemplars. Left (**a**): Selected Texas counties (black boundaries) with superimposed Thiessen polygons (red boundaries); MC = 0.051. Middle (**b**): The maximum NSA map pattern for the Texas counties surface partitioning; relatively speaking, green denotes $y_L$, yellow denotes $y_M$, and red denotes $y_H$. Right (**c**): Dot plot frequency distributions of the sets of eigenvalues for Figure 5a (top) and Figure 5b (bottom).

This eigenfunction context uncovers yet another, albeit more technically and algorithmically oriented, choice aspect of NSA. Estimating the spatial autocorrelation parameter of spatial autoregressive models requires the extreme eigenvalues of the spatial weights matrix, usually either **C** or **W**, to establish its feasible parameter space. The matrix powering algorithm $\mathbf{1}^T\mathbf{C}^{\tau+1}\mathbf{1}/\mathbf{1}^T\mathbf{C}^{\tau}\mathbf{1}$ almost always converges on the principal eigenvalue of matrix **C** as τ goes to infinity (its accompanying principal eigenvector is the normalized vector $\mathbf{E}_1$ converged upon by a normalized $\mathbf{C}^{\tau+1}\mathbf{1}$); the Perron-Frobenius theorem divulges that the principal eigenvalue of matrix **W** is 1 (its principal eigenvector is proportional to the vector **1**). The difficulty is determining the most extreme negative eigenvalue of these matrices. Griffith [16] devised an efficient algorithm that exploits NSA to estimate these eigenvalues based upon a Rayleigh quotient: An eigenvector is constructed that maximizes the differences between all neighboring elements, as defined by matrix **C** or **W** while being orthogonal to $\mathbf{E}_1$. Griffith et al. [17] show that it is more efficient than many of the standard computer software algorithms for calculating extreme eigenvalues of a matrix.

Another distinguishing facet of NSA connects to the special feature of negative bivariate correlation characterizing linear regression residuals. The mean of the sampling distribution of MC for normal random variables is E(MC) = −1/(n − 1), which is exactly the same as the conventional linear regression result for $E(e_i e_j)$. However, if the mean of variable Y is known, then E(MC) = [n/$\mathbf{1}^T\mathbf{C1}$] $E\sum_{i=1}^{n}(y_i - \mu)\sum_{j=1}^{n}c_{ij}(y_j - \mu)]/E[\sum_{i=1}^{n}(y_j - \mu)^2]$ = 0. In other words, the expected value of the MC differentiating between PSA and NSA is zero, not −1/(n − 1); this outcome of a negative expected

value has nothing to do with spatial autocorrelation, and everything to do with linear regression residuals. This negative quantity operates as a bias adjustment factor, similar to the multiplication of the maximum likelihood sample variance estimator, $s^2$, by n/(n − 1) to obtain an unbiased estimator. It arises because of the linear constraint placed on the calculation of MC by using the sample mean $\overline{y}$ when µ is unknown. Although this confusion does not alter the mechanics of linear regression residual spatial autocorrelation evaluations, it does alter the understanding of spatial autocorrelation. A better understanding of the NSA would have avoided this confusion.

An additional exclusive NSA feature is its tendency to occur as overshadowed correlation in a mixture with PSA. The moniker mixture, when attached to spatial autocorrelation, refers to some of the latent spatial correlation being PSA, and some being NSA, for a given geographic resolution and scale. It manifests because, for example, synchronization attributable to a common factor or spatial interaction amongst locations that produce PSA coincides with, for example, spatial competition or displacement that produces NSA. This context requires two spatial autocorrelation parameters in a spatial autoregressive model specification [18]. It suggests a violation of stationarity with regard to the nature of spatial autocorrelation being constant across a geographic landscape. Local indicators of spatial association (LISA; see reference [19]) furnish one tool for detecting this violation: Some of the individual polygon pair cross-product terms in the numerator of the global MC suggest global NSA, whereas others suggest global PSA, their magnitudes being beyond relatively small random fluctuations connected with zero spatial autocorrelation. This NSA masking may explain why spatial analysts rarely detect NSA. Model results failing to account for NSA when it is present in georeferenced data suffer from specification error. Experience to date suggests that in a mixture context (see Appendix A), this component usually accounts for no more than about 10% of the variation attributable to redundant information contained in variable Y. Although this appears to be a relatively small effect, many investigators employ a 5% rule, considering facets accounting for at least 5% of the variance in regression or principal components/factor analysis as attention worthy, and not small effects justifiably ignored.

In summary, NSA exhibits at least six prominent special advantageous insights not furnished by PSA alone. Foremost is that it cannot manifest itself the same way in both discrete and continuous geographic space, recurrently being a rarely detected characteristic of polygonal data. NSA links to spatial competition, and as such offers the potential of serving as a geographic compactness index. The MC, the most commonly employed spatial autocorrelation index, has a positive benchmark endpoint that usually extends to 1.2 or more. In contrast, the NSA benchmark endpoint for the MC frequently is closer to −0.5. As such, MC measures of PSA tend to appear stronger than they actually are, whereas measures of NSA tend to appear weaker than they actually are. Meanwhile, extreme NSA can be exploited to efficiently and effectively calculate the extreme eigenvalues of certain matrices. In addition, the boundary between PSA and NSA is zero, not −1/(n − 1), which is a bias adjustment factor resulting from a linear constraint placed on the calculation of the sample MC. Finally, NSA often accompanies PSA as subordinate correlation in a mixture of the two.

## 2. A Brief Overview of Moran Eigenvector Spatial Filtering (MESF)

The preceding section mentions eigenfunctions of matrix expression (2), the matrix version of part of the numerator of the MC defined by Equation (1), with reference to NSA. It also notes that certain auto-probability functions (e.g., auto-Poisson) cannot accommodate one or the other of the two natures of spatial autocorrelation. Furthermore, the auto-normal based spatial autoregressive models require nonlinear regression, whereas generalized linear models (GLMs) require Markov chain Monte Carlo (MCMC) techniques, for parameter estimation. These and other numerically intensive complications (e.g., determining the eigenvalues of massively large georeferenced datasets) inspired the development of alternative spatial statistical methodologies. Griffith and Chun [20] furnish one of a number of available overviews of MESF, an innovative spatial statistical methodology that is an alternative to spatial autoregression, and that adds a set of synthetic proxy variables—eigenvectors extracted from

matrix expression (2), an adjusted spatial weights matrix that describes connections among geographic objects in space—as control variables to filter spatial autocorrelation out of regression residuals and transfer it to the mean response in a model specification; the intercept term changes from a constant to a variable, like with mixed model specifications. These control variables detect and isolate stochastic spatial dependencies among georeferenced observations, thus allowing model parameter estimation to proceed with observations mimicking being independent.

MESF, whose seminal publication appeared in 1996, is in the tradition of dimension reduction analysis (see reference [21]). It was also discovered independently several years later by Borcard and Legendre [22] as principal coordinate analysis of neighbor matrices (PCNM). PCNM computes the eigenfunctions of a matrix containing truncated pairwise geographic distances between locations, alluding to more of a geostatistical perspective. Griffith [23] presents the basic theory of MESF. Griffith and Peres-Neto [24] demonstrate the general equivalence of MESF and PCNM. Describing these statistical methodologies in terms analogous to the multivariate statistical technique of principal components analysis (PCA) helps to explain both of them. PCA deals with a correlation matrix, **R**, that has all ones in its diagonal entries, and all off-diagonal entries that are less than or equal to one in absolute value. Consequently, its eigenvalues are non-negative. A spatial weights matrix **C** has all zeroes in its diagonal entries, with a sizeable number of zeroes in its off-diagonal entries [planar graph based polygon adjacency definitions result in no more than $6(n-3)$ non-zero entries; distance truncations usually result in far less than 50% non-zero entries]. Consequently, its eigenvalues are both positive and negative (see Figure 5c). MESF and PCNM treat a modified version of matrix **C** (see Equation (2)) like PCA treats matrix **R**. Eigenfunction calculations are for these specific matrices. PCA uncovers dimensions spanning the variables used to construct matrix **R**. It computes scores by constructing linear combinations of the original variables, with the coefficients of a given linear combination being the elements of a particular eigenvector; sometimes this synthetic variate is multiplied by the square root of its eigenvector's corresponding eigenvalue. Matrix **R** eigenvalues represent variance accounted for by each principal component; this multiplication seeks to preserve the original total variance. These synthetic variates are orthogonal and uncorrelated. Because the square root of negative eigenvalues is a complex number, PCNM initially and erroneously dismissed their eigenfunctions. However, the eigenfunctions of matrix expression (2) pertain to spatial autocorrelation, with negative eigenvalues indexing NSA. The eigenvectors here are also orthogonal and uncorrelated. In contrast to PCA, for MESF and PCNM, the eigenvectors themselves are synthetic variates; they are not used to construct linear combinations. Each element of a given eigenvector links to a particular location, in accordance with the ordering defined by the affiliated spatial weights matrix. Therefore, eigenvectors can be mapped (see Figure 5b), visualizing the nature and degree of spatial autocorrelation designated by their corresponding eigenvalues. In both PCA and MESF/PCNM, the synthetic variates produced can be included as covariates in a regression equation. By doing so, MESF eigenvectors account for residual spatial autocorrelation.

The crucial matrix algebra quantities extracted from an adjusted spatial weights matrix **C** are the eigenfunctions of matrix expression (2), which appears in the numerator of the MC. Eigenvalues, $\lambda$, are the n scalar solutions to the nth-order polynomial matrix equation

$$\det[(\mathbf{I} - \mathbf{1}\mathbf{1}^\mathrm{T}/n)\mathbf{C}(\mathbf{I} - \mathbf{1}\mathbf{1}^\mathrm{T}/n) - \lambda\mathbf{I}] = 0;$$

the corresponding eigenvectors **E** are the non-trivial vector solutions (i.e., $\mathbf{E} \neq \mathbf{0}$) to the equation

$$[(\mathbf{I} - \mathbf{1}\mathbf{1}^\mathrm{T}/n)\mathbf{C}(\mathbf{I} - \mathbf{1}\mathbf{1}^\mathrm{T}/n) - \lambda\mathbf{I}]\mathbf{E} = \mathbf{0}.$$

These eigenfunctions are the basis of MESF and are the synthetic variates that can account for non-zero spatial autocorrelation in regression residuals. Their linear combination is an eigenvector spatial filter (ESF). Appealing additional properties of MESF eigenvectors include: (1) One vector is proportional to the vector **1**, the intercept covariate in a regression model specification; and, (2) this

conceptualization relates to the spatially structured random effects component of mixed models [25]. Including eigenvectors as covariates, by selecting relevant ones with a stepwise procedure (see Appendix A), enables spatial autocorrelation to be accounted for in a conventional statistical estimation context, in either a linear or a GLM specification.

In conclusion, the eigenfunctions of matrix expression (2) support various revelations about NSA. When used to construct an ESF, they allow Poisson and NB regression with georeferenced data without concern for the nature of spatial autocorrelation (i.e., they circumvent the NSA-only restriction of certain auto-probability models). They furnish a simple way to detect mixtures of PSA and NSA, allowing the construction of two ESFs, a rather complicated estimation problem for spatial autoregressive specifications (see reference [18]). They also permit the assessment of specification error when NSA is overlooked in, and hence excluded from, a model specification. The ensuing illustrative data analyses employ MESF to handle spatial autocorrelation.

## 3. Selected Case Studies Demonstrating the Presence and Importance of NSA

NSA most often naturally materializes with competitive and displacement locational processes, negative spatial externalities, the construction of spatial correlograms, the spectrum (i.e., eigenvalues) of a spatial weights matrix, the calculation of linear and nonlinear regression residuals, and the computation of LISA, to enumerate some of its possible contexts. However, many georeferenced attribute variables fail to measure these types of phenomena, to the detriment of NSA. A standard development trajectory for a concept begins with its establishment. NSA already has this concept formulation in the abstract. The second step is quantification. Although indices exist for calculating the nature and degree of NSA (e.g., the MC and the Geary Ratio), preceding discussion argues for the establishment of a more intuitive and informative yardstick, one that spans the interval $[-1, 0)$; mathematical statistical distribution theory work will need to accompany this development. The next step is hypothesis testing (e.g., $H_0$: zero spatial autocorrelation), which depends upon variables studied. Scientific orthodoxy and societal "big bang for the bucks" viewpoints sometimes stifle, and even eliminate, the types of variables studied, at least until expert opinion governing these paradigms shifts. Over the years, the former has placed far more emphasis on cooperation and expansion, whereas the latter has deemed many static or comparative static investigations fashionable. However, some change in variable subject matter is underway, such as defining the population of areas in terms of the time of day rather than permanent residence a la census reports (i.e., differences between daytime and nighttime population in a location). The final step for concept development is model identification, specification, and estimation (e.g., spatial autoregressive, geostatistical semi-variogram, and MESF equations). These four steps form a natural progression, from concept formation, through the univariate statistical description and sample-to-population inference, to formal model representation.

This section summarizes four specimen analyses in order to highlight important but seemingly unnoticed roles of NSA. The first addresses quantification, the second step in concept development. The second and third address variable definitions as well as mixtures of PSA and NSA, the third step in concept development. These PSA–NSA mixtures allude to issues about stationarity, ones that in part LISA can illuminate. Juxtaposed contrasting polygon attribute values merit attention, even if their numbers are considerably fewer than those of their counterparts for PSA. Finally, the fourth addresses modeling and specification error, the fourth step in concept development. Understanding NSA in these examples is essential because it is an important descriptor of their data, just like the mean, variance, frequency distribution, and PSA, and it furnishes insights about spatial competition, areal unit polygon formation, trade-offs between PSA and NSA, especially regarding stationarity, and regression model misspecification.

### 3.1. Market Area Competition: NSA and Facility Closures

The example of geographic competition in terms of polygon-region area changes can be extended to the context of geographic retail market area competition. Multi-locational chain store closures

present one instance of a spatial competition process. This topic is particularly relevant because of the widespread and large number of multi-locational retail firms closing their brick-and-mortar stores. Many such companies have been liquidating large numbers of stores for several years now, including (based upon newspaper reports during the last five years): Abercrombie & Fitch, Aeropostal, American Apparel, American Eagle Outfitters, Barnes & Noble, BCBG, Bebe Stores Inc., Children's Place, Crocs, Family Dollar, GameStop, HHGregg, JC Penny, Kemp Mill Music, Kenneth Cole, Macy's, Office Depot, Payless ShoeSource, Radio Shack, Sears and Kmart, Sports Authority, Staples, The Limited, the United States (US) Postal Service, and Wet Seal. Two additional chains examined in this section are Kohl's department stores and Albertsons grocery stores within the Dallas-Forth Worth (DFW) Metropolitan Statistical Area (MSA). The small sample sizes involved here should not be confused with the selected closure problems being too simplistic; these are the actual numbers of stores in the DFW MSA, and substantial mileage separates this local network from the next closest local network of stores, isolating them from the additional peripheral geographic competition. Although results summarized in this section are illustrative, they could be extended to retail store chains, such as Walmart, that have far more comprehensive geographic coverage.

Individual store location market areas can be approximated with Thiessen polygons. When a store closes, its market area shrinks to zero, and some or all of its surrounding nearby chain outlets experience an increase in their market areas. The ratio of the aggregate population captured by the paired Thiessen polygons in these two geographic distributions reflects a spatial competition effect; a closed store's ratio becomes 0, ratios for stores that cannibalize the closed store's market area become greater than 1, and ratios for the remaining stores are 1. Consider the Kohl's department store chain, and the Albertsons grocery store chain, in the DFW MSA. Approximating the market areas of individual stores using Thiessen polygons (i.e., the assumption that people patronize the closest store of a chain; see Figure A1, Appendix B) can capture market area size by summing all 2016 census tract populations whose centroids are located within each Thiessen polygon. In recent years, Kohl's did a single store closing, and Albertsons closed eight stores. These closures caused a redistribution of some market areas (Figure 6).

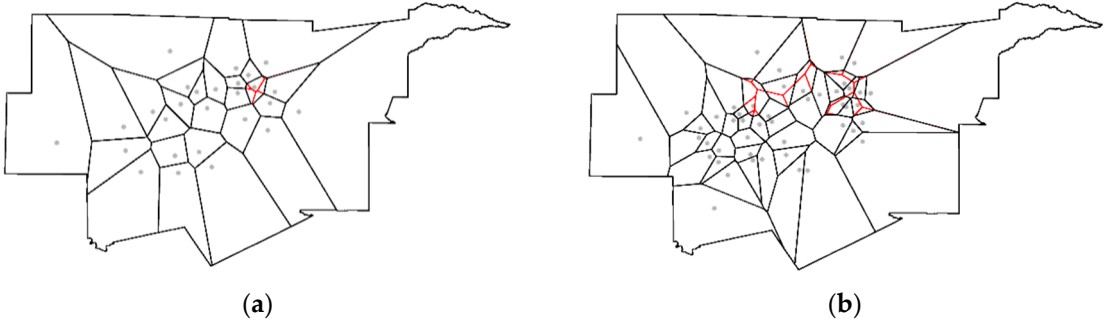

|       |       |
| :---: | :---: |
| (**a**) | (**b**) |

**Figure 6.** Approximate market areas (boundaries denoted by Thiessen polygons) for Kohl's and Albertsons stores in the Dallas-Forth Worth Metropolitan Statistical Area (DFW MSA), with market area redistributions denoted by red lines, and open stores denoted by gray circles. Left (**a**): Kohl's department stores. Right (**b**): Albertsons grocery stores.

The Kohl's department stores case furnishes proof of concept, demonstrating the feasibility of NSA emerging in a real world geographic landscape. Even with a single store closure (in a network of 27 stores), NSA (most spatial autocorrelation appears to be some type of PSA–NSA mixture, and hence, spatial autocorrelation mixtures essentially are present in many, if not most, georeferenced data; this notion is the topic of §3.2–3.4 and their examples; only the NSA component of spatial autocorrelation is the focal point of the other examples discussed here, because presently a need exists to establish its presence) occurs [MC = −0.080 < −1/(27 − 1)]; however, although indicative of NSA (i.e., it is less than −0.04), this MC value was not statistically significant. Nevertheless, it did

demonstrate in a non-trivial way that even a single closure induced NSA. Figure 7a portrays the Moran scatterplot for this case, which unmistakably has a negative sloping trend line. Based on the MC, the maximum NSA possible for the Figure 6a configuration of polygons was −0.525, which, stretching the endpoint of the MC scale to −1, suggested that −0.080 should be viewed more like a value of −0.152 (≈ −0.080/|−0.525|). Meanwhile, with eight of 47 stores closed in the Albertsons grocery stores case, NSA was very conspicuous [MC = −0.218, which was statistically significant (z = −2.2)]. Figure 7b portrays the Moran scatterplot for this case, which clearly had a negative sloping trend line. Based on the MC, the maximum NSA possible for the Figure 6b configuration of polygons was −0.525, which, again stretching the endpoint of the MC scale to −1, suggested that −0.218 should be viewed more like a value of −0.415 (≈ −0.218/|−0.525|).

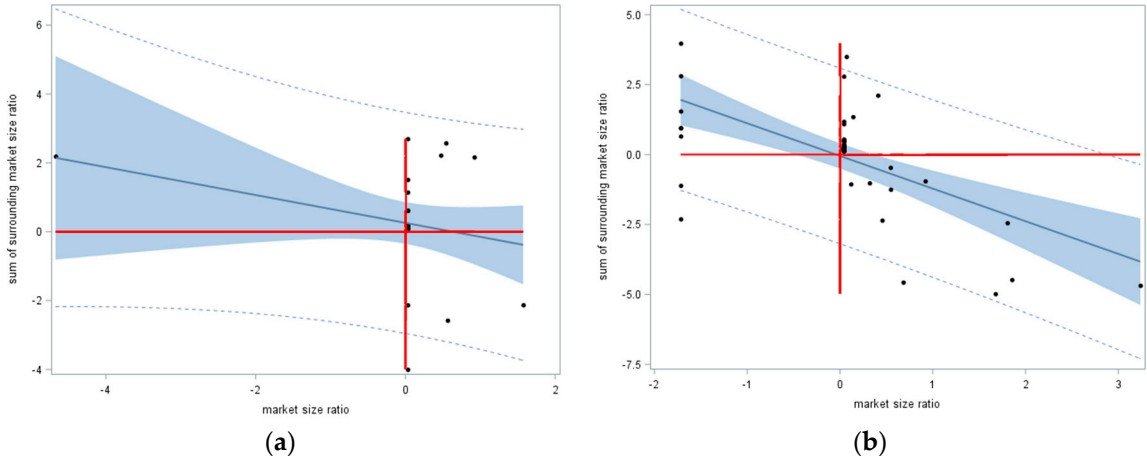

**Figure 7.** z-score axes Moran scatterplots for market area population ratios (after-closure size divided by before-closure size). Left (**a**): Kohl's department stores. Right (**b**): Albertsons grocery stores.

The aforementioned Texas area competition example furnishes a situation in which NSA measured by the MC can be used to index compactness. The market area examples presented in this section furnish a situation in which NSA as measured by the MC in leave-one-out (e.g., a jackknifing or cross-validation style of the procedure) hypothetical store closings can be used to index impacts of store closures. In other words, calculating the MC for each store's market area set to zero, in turn, yields an MC as an index that can be related to other store attributes. Figure 8a portrays the geographic distribution of the 26 Kohl's stores that are currently open. Figure 8b portrays the corresponding 26 Moran scatterplot trend lines for the leave-one-out market size reallocations, and Table 2 reports their accompanying MC values.

Table 2 tabulates the MC values as well as their adjustments for stretching the endpoint of the MC scale to −1; these latter indices suggest that, on average, the NSA is weak-to-moderate in degree. Employing a liberal benchmark of α = 0.10, only six values were not statistically significant. Meanwhile, Figure 8a reveals that the polygonal market areas were larger in area along the periphery encircling the DFW MSA, where population density was lower. Normal quantile plots furnished no evidence to indicate that either the 26 MC values or either of their core-periphery subsets of 10 and 16 MC values did not conform to a normal distribution. Interestingly, an MESF analysis found that the geographic distribution of the 26 NSA MC values displayed NSA, which accounted for about 10% of its geographic variation. Even after adjusting for this NSA, the difference between the average NSA MC values for the core and periphery regions was not statistically significant. Consequently, the only trend in these MC values was for larger NSA index values to be neighbors of smaller NSA index values. Nevertheless, these MC values disclosed that closing Store #17 would have the biggest competitive impact, whereas closing Store #15 would have the smallest impact, upon its surrounding market areas.

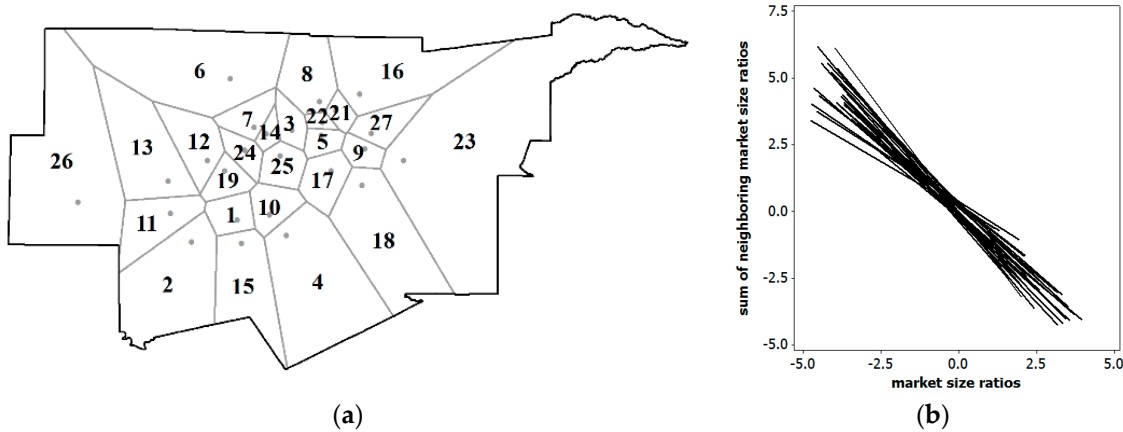

**Figure 8.** The 26 Kohl's open department stores in the DFW MSA. Left (**a**): Locations and market areas. Right (**b**): The 26 NSA Moran scatterplot trend lines using z-score axes, each involving a hypothetical single store closure.

**Table 2.** MCs for a leave-one-out (i.e., a single store closure) market size competition analysis for Kohl's department stores in the DFW MSA.

| Store ID | MC | MC/|MC$_{min}$| | Prob (H$_0$) | Store ID | MC | MC/|MC$_{min}$| | Prob (H$_0$) |
|---|---|---|---|---|---|---|---|
| 1 | −0.206 | −0.383 | 0.091 | 14 | −0.181 | −0.336 | 0.129 |
| 2 | −0.250 | −0.465 | 0.045 | 15 | −0.136 | −0.253 | 0.219 |
| 3 | −0.280 | −0.520 | 0.027 | 16 | −0.205 | −0.381 | 0.092 |
| 4 | −0.225 | −0.418 | 0.068 | 17 | −0.311 | −0.578 | 0.014 |
| 5 | −0.195 | −0.362 | 0.106 | 18 | −0.243 | −0.452 | 0.051 |
| 6 | −0.213 | −0.396 | 0.082 | 19 | −0.230 | −0.428 | 0.062 |
| 7 | −0.215 | −0.400 | 0.079 | 21 | −0.250 | −0.465 | 0.045 |
| 8 | −0.217 | −0.403 | 0.077 | 22 | −0.159 | −0.296 | 0.170 |
| 9 | −0.236 | −0.439 | 0.057 | 23 | −0.233 | −0.433 | 0.060 |
| 10 | −0.269 | −0.500 | 0.033 | 24 | −0.202 | −0.375 | 0.095 |
| 11 | −0.284 | −0.528 | 0.024 | 25 | −0.276 | −0.513 | 0.028 |
| 12 | −0.200 | −0.372 | 0.098 | 26 | −0.148 | −0.275 | 0.192 |
| 13 | −0.252 | −0.468 | 0.044 | 27 | −0.183 | −0.340 | 0.125 |

NOTE: Store #20 is the one that already closed; MC$_{max}$ = 0.855, and MC$_{min}$ = −0.538 (this value differs from that for Figure 6 because n is different).

In conclusion, in the context of quantification, NSA shows promise as a tool for assessing compactness of polygons, which could prove useful for such endeavors as political redistricting map assessments. This same conceptualization can be extended to measuring market area change in the presence of closures, another quantification context; this NSA outcome was detectable even with the closure of only a single facility (in the Kohl's example, some MCs were not significant, but all were negative, a finding endorsing that it is not a trivial example). With the large number of multi-location chains undergoing contraction/rationalization via closures today—even Subway announced in April of 2018 that it plans to close 500 more US stores, after already closing 800 stores in 2017—encountering NSA in geospatial data analyses should be a more frequent occurrence. Furthermore, relating stores' MC values to their business characteristics has the potential to help decision makers with closure site selections.

### 3.2. Journey-To-Work: Shifts In Daytime and Nighttime Populations

The NSA theme here is twofold: One of variable definition, and the other of a PSA-NSA mixture. Historically, denominators of rates and populations at risk relied on georeferenced population data for people geotagged to their locations of permanent residence (i.e., nighttime population). The Schelling [26,27] model indicates that comparatively homogeneous residential areas cluster in

geographic space. Urban economics land use theory indicates that commercial and other economic activities locationally organize in particular clustered ways in geographic space. Government zoning ordinances reinforce this overall spatial organization, separating, for example, most residential and employment areas in a geographic landscape; some, but not considerably much blending tends to occur, such as overlapping retail and residential land uses. This geographic composition tends to generate moderate PSA. However, daily population flows—e.g., the journey to work, journey to shops, and journey to recreate—redistribute population over a geographic landscape during different times of a day. This redistribution suggests the presence of NSA, if, for example, the variable of interest is the ratio of daytime and nighttime populations.

The 2000 journey-to-work flows in the DFW MSA exemplify this general situation. In this illustration, the total population of this MSA was 5,161,544, the number of workers traveling from census tracts to their jobs was 2,461,523 (including within census tract travel), and the number of workers traveling to census tracts to their jobs was 2,499,569 (including within census tract travel); these counts included workers going into and out of the MSA from other places (which was why these two sums differ). Figure 9 portrays the geographic distribution of a Box-Cox transformed daytime-to-nighttime population ratio (the employed Box-Cox power transformation involves an exponent of −1.25, and a translation parameter of zero: $Y^{-1.25}$; for simplicity, the inverse itself, $1/Y$, also was explored; in either case, because the translation parameter is zero, the variable is equivalent to a nighttime-to-daytime population ratio with an exponent of 1.25).

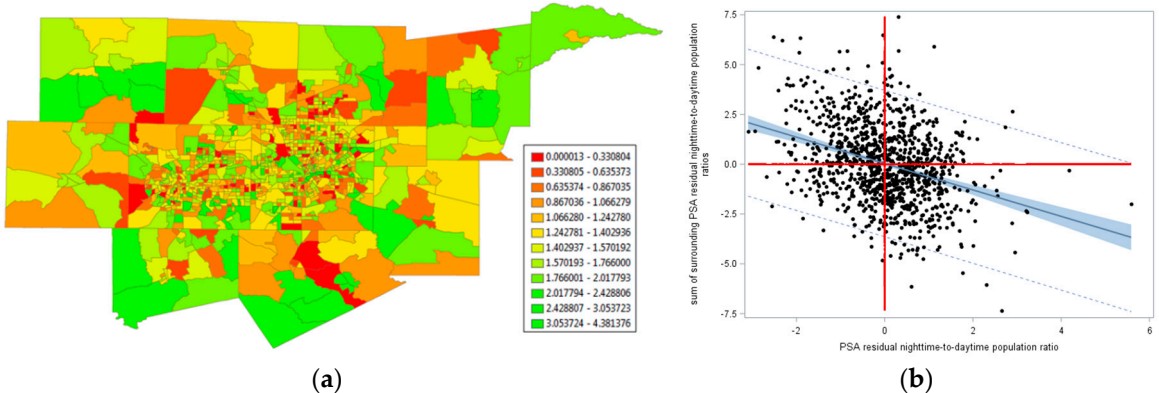

(**a**)  (**b**)

**Figure 9.** A 2000 daytime-to-nighttime population ratio for DFW. Left (**a**): The geographic distribution of the Box-Cox transformed ratio (relatively speaking, green denotes $y_L$, yellow denotes $y_M$, and red denotes $y_H$). Right (**b**): A z-score axes Moran scatterplot for this transformed ratio.

NSA was conspicuous in Figure 9a (i.e., many extremely high and extremely low values were juxtaposed), with MC = 0.072, implying no spatial autocorrelation (or, perhaps, trivial PSA). However, this situation was a mixture of PSA and NSA. Accounting for the PSA in the geographic distribution of the ratio studied here with an MESF analysis—which statistically explained about a quarter of the geographic variance in this phenomenon—produced linear regression residuals whose Moran scatterplot is portrayed in Figure 9b. These residuals contained statistically significant NSA ($z_{MC}$ = −3.8), with MC = −0.123. Because its lower limit, in this case, was −0.730, stretching its MC measurement scale endpoint to −1 indicated that this NSA was stronger than it appears (i.e., −0.168 ≈ −0.123/|−0.730|; meanwhile, the positive scale stretched to 1.179, indicating that 0.072 effectively was more like 0.061 ≈ 0.072/1.179, a value closer to the expected value of −1/(1046−1) ≈ −0.001). An MESF analysis revealed that NSA accounted for nearly 20% of the geographic variation in this Box-Cox transformed daytime-to-nighttime ratio.

Figure 4a implies that the boundaries of polygons play a role in detecting NSA. If these boundaries are delineated such that polygons internalize PSA, then a PSA–NSA mixture should (nearly) disappear, and what remains should be NSA. In other words, for the DFW example, a merging of nearby

similar values in the numerous small clusters (PSA) should allow NSA to be more prominent. Viewed slightly differently, judicious polygon aggregation enables an exploration of the possibility of restoring stationarity to spatial autocorrelation when a mixture of PSA and NSA occurs. A cluster analysis was performed to achieve this end: Adjacent census tracts with either $y_H$ or $y_L$ neighboring values were merged (this implementation was inspired by Traun and Loidl [28]). The outcome was a reduction in n from 1046 to 472 (this change of geographic resolution is comparable to using ZIP (zone improvement plan) code tabulation areas rather than census tracts). Adjacent census tracts with relatively similar Box-Cox transformed daytime-to-nighttime population ratios were merged (Figure 10a), which eliminated a major source of PSA when calculating MC values; it essentially constituted the PSA component of the prevailing PSA–NSA mixture. The resulting geographic distribution exhibited starker contrasts (Figure 10b) and yielded a Moran scatterplot (Figure 10c) portraying NSA very similar to that for the PSA-adjusted residuals used to construct Figure 9b. Now MC = −0.123 ($z_{MC}$ = −4.7). Because the minimum MC value (the original census tract surface partitioning has extreme MC values of −0.730 and 1.179, which is very similar to those for the aggregated census tract surface partitioning of −0.723 and 1.191) here was −0.723, this NSA index value was more like one of −0.185 ($\approx$ −0.123/|−0.723|) when stretching the endpoint of its MC scale to −1. An MESF analysis revealed that PSA accounted for roughly 9% of the geographic variation in this spatially aggregated Box-Cox transformed daytime-to-nighttime ratio, a substantial decrease from about 25%. This MESF analysis also revealed that NSA accounted for 32% of this geographic variation, a substantial increase from nearly 20%. The new weak mixture ESF residuals contained a marginal amount of spatial autocorrelation, producing $z_{MC}$ = −1.97 (the expected value was 0.011, very close to zero); it was barely statistically but not substantively significant, given its expected value was thus close to zero Therefore, this data experiment confirmed that accounting for PSA through polygon merging removes its masking of NSA. It also confirmed that the role geographic resolution, polygon and variable definitions, and PSA–NSA mixtures played in detecting NSA can be crucial.

In conclusion, more NSA cases may be detected in the future because of a change in studied variables, some of which were never contemplated before. In addition, more customized geographic aggregation of individual geocoded data may well produce polygon delineations that render NSA in resulting aggregate geographic distributions. Polygon construction (e.g., census tracts, municipal districts) should not camouflage NSA by always averaging it with PSA; intentionally highlighting it may be important for certain contrasts. Regardless, most likely PSA–NSA mixtures will be uncovered in many more geographic distributions of phenomena. A principal implication here is that rates and population-at-risk quantities need to reflect the time of day associated with counts of events.

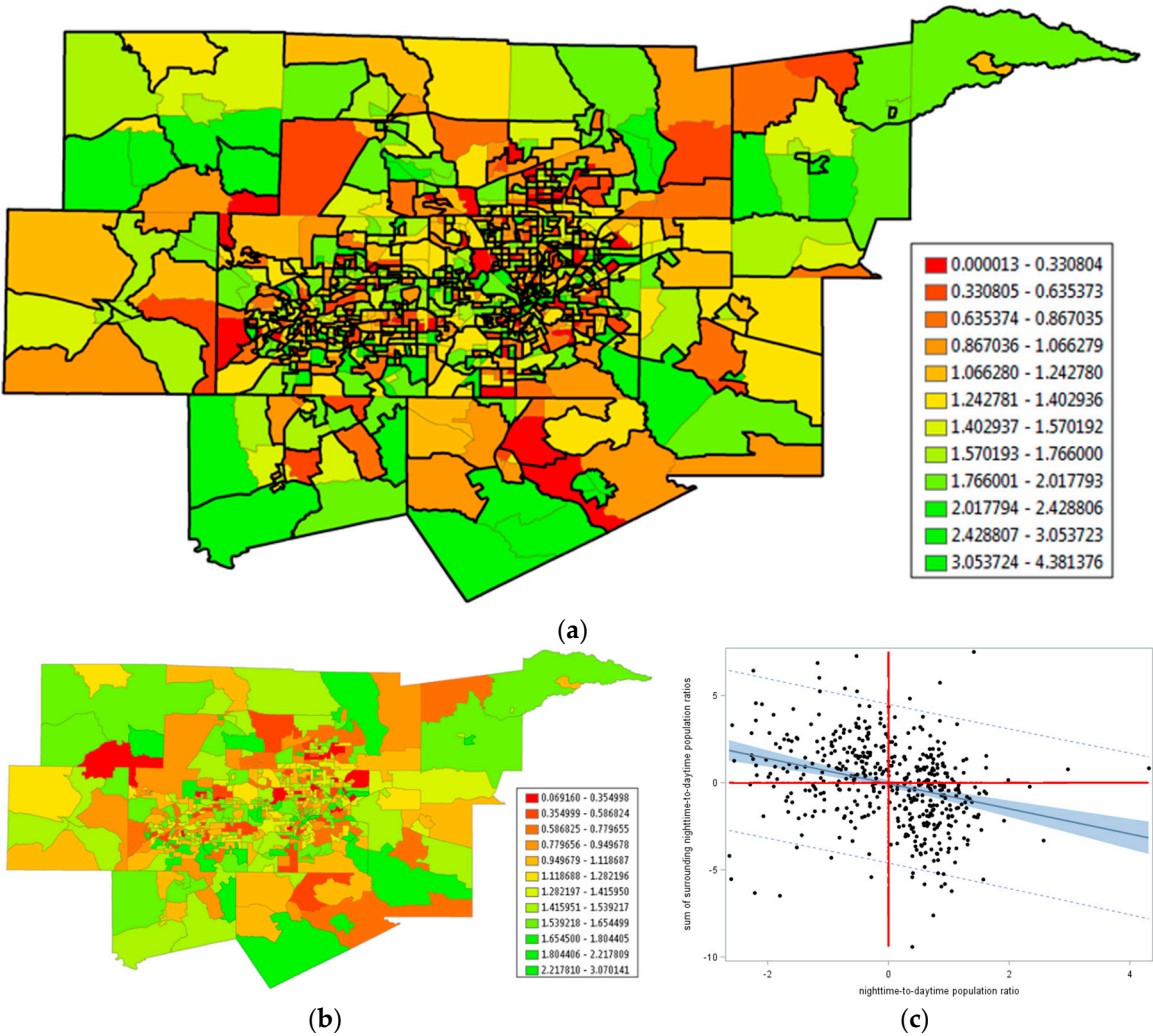

**Figure 10.** An aggregated 2000 inverse daytime-to-nighttime population ratio for DFW; relatively speaking, green denotes $y_L$, yellow denotes $y_M$, and red denotes $y_H$. Top (**a**): An overlay of the aggregated areal unit boundaries on the original census tract resolution geographic distribution. Bottom left (**b**): The geographic distribution of the geographically aggregated data ratio. Bottom right (**c**): A z-score axes Moran scatterplot for the aggregated data ratio.

## 3.3. Urban Area Shrinkage

Shrinking cities, which are numerous, afford another previously seldom considered empirical landscape in which NSA naturally materializes. Weaver et al.'s [29] list of such cities included Detroit, which had a population decrease of roughly 25% between the 2000 and 2010, and roughly 60% between the 1950 and 2010, US decennial censuses. One local government response in Detroit has been to discourage people from living in some parts of the city, and encouraging them to live in other parts, in order to rearrange residents into preferred clusters of higher population density. NSA is a natural outcome of this type of geographic realignment (a style of locational displacement). A visual inspection of Figure 11a, which portrays the ratio of 2010 to 2000 Detroit population by census tract, corroborates this contention. For this untransformed variable case, MC = $-0.025 < -1/(308-1)$; MC detects slight but non-significant NSA in this geographic distribution of population density, which has an extremely skewed frequency distribution.

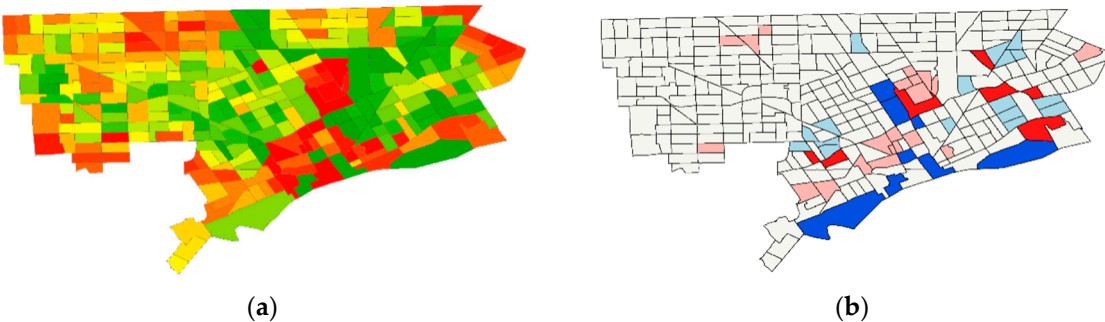

<center>(**a**)                                                        (**b**)</center>

**Figure 11.** Detroit. Left (**a**): The geographic distribution of the population ratio across census tracts; relatively speaking, green denotes $y_L$, yellow denotes $y_M$, and red denotes $y_H$. Right (**b**): A Local indicators of spatial association (LISA) statistics map; dark red is high-low (H–L), dark blue is low-high (L–H), light red is high-high (H–H), and light blue is low-low (L–L).

Because the population ratios are highly non-normal, they were subjected to a Manly transformation (an extension of the Box-Cox transformation to exponentiation), yielding $1 - 1/e^{ratio}$; this transformation increases the Shapiro-Wilk (S–W) normality diagnostic statistic from 0.16 to 0.85 (it remains significant, but is much closer to 1, as well as indicative of better symmetry) (Manly [30] argues for adding $e^{\alpha y}$ to the family of Box-Cox transformations; here the estimate of $\alpha$ was approximately $-1$; because this inverse transformation changed the nature of the relationship between Y and other variables, it was subtracted from 1 to preserve these original relationships).

Scatterplots such as Figure 12a show deviations from the trend line, which has a slope related to the MC. In the tradition of linear regression residual analysis, these deviations can be the subjects of local analyses. This outlook spawned the development of LISA and other local spatial autocorrelation statistics. These statistics uncover the presence of heterogeneity in a global spatial autocorrelation index, part of which can be a mixture of PSA and NSA. Figure 11b portrays this situation for the 2000–2010 Detroit population density data. The previous description of PSA trend in a Moran scatterplot is in terms of $y_H$ tending to pair with $y_H$, in the first quadrant of the scatterplot, and $y_L$ tending to pair with $y_L$, in the third quadrant of the scatterplot. LISA respectively denotes these pairings as H–H and L–L. In addition, the previous description of NSA trend in a Moran scatterplot is in terms of $y_L$ tending to pair with $y_H$, in the second quadrant of the scatterplot, and $y_H$ tending to pair with $y_L$, in the fourth quadrant of the scatterplot. LISA respectively denotes these pairings as L–H and H–L. Figure 11b portrays the LISA statistics map for Detroit, which highlights the prevalence of H–L and L–H population change ratio clusters. Supplementing these pockets of contrast are clusters of similar values, implying the presence of a PSA–NSA mixture geographic distribution. For this case, $MC/MC_{max} = 0.135/1.12493 \approx 0.120$, indicating very weak global PSA.

An inspection of Figure 12a unveils that a number of individual MC covariation terms, some indicating PSA and some indicating NSA, lie outside of the global 95% prediction interval. Figure 12b unveils that these as well as some that lie inside but close to the 95% prediction interval boundaries, together with a few that might be considered leverage points, constituted the significant LISA; 40 statistically significant LISA indicated PSA, and 14 statistically significant LISA indicated NSA. This particular mixture of cross-product terms implied nonstationary spatial autocorrelation across the studied geographic landscape. Figure 12b also visualized the trend lines based upon these more extreme LISA. An MESF analysis showed that PSA accounted for roughly 33% of the geographic variance in the Box-Cox transformed 2000–2010 Detroit population change ratio. The MC calculated with its residuals was $-0.197$, which effectively was $-0.289$ ($\approx -0.197/|-0.682|$) after stretching the MC measurement scale to $-1$. Figure 12c portrays its Moran scatterplot, confirming the presence of NSA, which accounts for an additional roughly 25% of the geographic variance. The mixture ESF residuals contained only a trace amount of spatial autocorrelation, producing $z_{MC} = 0.56$ (the expected value was $-0.019$, very close to zero).

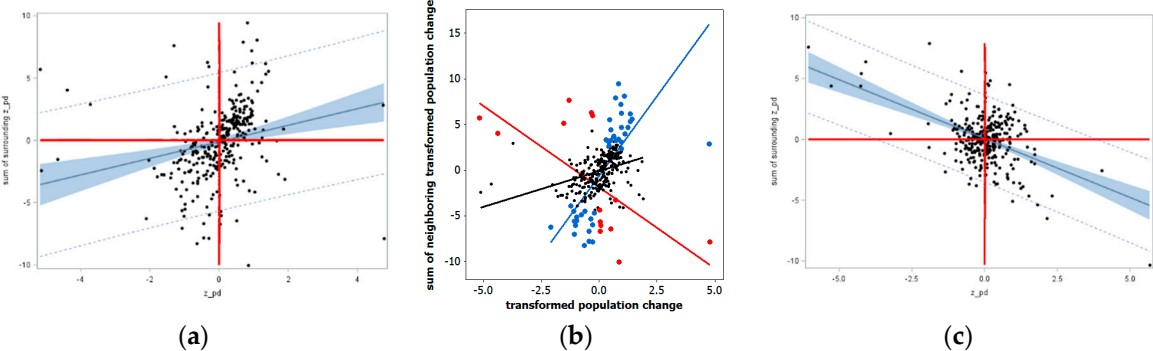

**Figure 12.** z-score axes Moran scatterplots for the 2000–2010 Detroit population change ratio. Left (**a**): Differentiating individual MC covariation terms according to 95% prediction (dotted line) and confidence limits (blue shaded area). Middle (**b**): Moran scatterplot with statistically significant LISA statistics identified (blue denotes H–H and L–L; red denotes L–H and H–L), with positive spatial autocorrelation (PSA) (blue), NSA (red), and global (black) trend lines superimposed. Right (**c**): The residual NSA scatterplot after adjusting for PSA.

In conclusion, the Detroit population change analysis uncovered at least local NSA with considerable evidence that it was part of a PSA–NSA mixture that accounted for a sizeable amount of geographic variation in the response variable Y. As implied by the preceding empirical example, most likely the PSA–NSA mixtures will be uncovered in many more spatial distributions of georeferenced phenomena. The principal implications here are that LISA can be indicative of PSA–NSA mixtures and that the contraction of urban spaces involves NSA, which requires appropriately defined variables to detect it.

### 3.4. 1990. Homicide Rates in the US South Revisited

County homicide data for the US south ($n = 1412$), analyzed by Baller et al. [31], furnish a PSA–NSA mixture example relating to the substantive conceptual expectation of locational displacement and illustrates specification error attributable to overlooking NSA. Part of the goal of Baller et al.'s paper was to select an appropriate spatial autoregressive specification describing county level homicide data for the US south. Baller and his collaborators use five covariates to account for socio-economic/demographic factors associated with these homicide counts, utilizing the three-year aggregate of homicide incidents between 1989 and 1991 as the response variable. Their analysis employs a normal approximation; the treatment of the homicides in this section is as counts, allowing the use of a GLM. Baller et al.'s spatial lag (i.e., spatial autoregressive response) model specification produces $\hat{\rho} = 0.230$ and a pseudo-$R^2$ of 0.333 for the three-year average homicide rates. The analysis summarized here made use of only the 1990 data, and employed a Box-Cox power transformation to better align the empirical homicide rates with a normal distribution. This revised analysis yielded $\hat{\rho} = 0.259$ and a pseudo-$R^2$ of 0.357. Thus, these two sets of results were compatible.

The presence of overdispersion (i.e., extra-Poisson variation) implied the need to employ a NB model specification with log-population as an offset variable. The covariates-only model produced a pseudo-$R^2$ of 0.311 with quasi-likelihood Poisson regression, and 0.308 with NB regression, which produced a dispersion parameter estimate of 0.106. One expectation was that the spatial autocorrelation latent in these data was a PSA–NSA mixture. A substantive rationale for this contention was the common place features in which many homicides occurred (PSA), on the one hand, and the crime displacement hypothesis (NSA), on the other hand. The analysis summarized here employed MESF because the auto-Poisson model specification was unable to accommodate PSA [13]. Of the 1412 adjusted spatial weights matrix eigenvectors [see matrix expression (2)], the posited candidate set included 352 containing PSA (based on $MC_j/MC_{max} \geq 0.25$), and 482 containing NSA (based on $MC_j/MC_{max} < -0.25$). The constructed ESF contained 110 PSA (ESF$_{PSA}$) and 102 NSA (ESF$_{NSA}$)

eigenvectors. The $ESF_{PSA}$ component (Figure 9a) accounted for roughly 17% of the geographic variation in homicide rates across the US south, and represented moderate PSA (MC = 0.735/1.111 ≈ 0.662), whereas the $ESF_{NSA}$ component (Figure 13b) accounted for roughly 12%, of this geographic variation, and represented moderate NSA (MC = –0.385/|–0.605| ≈ –0.636). The final pseudo-$R^2$ was 0.596, substantially greater than that for the normal approximation spatial lag specification. In addition, the inclusion of the PSA eigenvectors reduced the NB dispersion parameter from 0.106 to 0.026. Inclusion of the NSA eigenvectors reduced it to 0, returning the specification to a more parsimonious standard Poisson one. Some small amount of residual PSA existed, with MC = 0.018. The exact distribution theory for this residual spatial autocorrelation presently is unknown; hence, a simulation experiment (10,000 replications) was conducted to estimate its mean and variance. The simulation experiment rendered a $\overline{\text{pseudo} - R^2}$ of 0.660, slightly better than the observed 0.596. The test statistic was $z_{MC}$ = 1.72, which was neither statistically (two-tailed test with α = 0.10) nor substantively significant; its expected value of –0.014 was very close to zero.

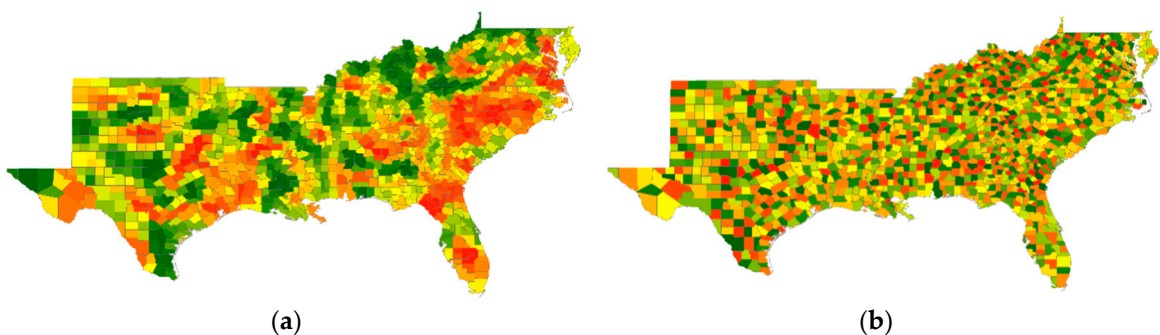

|  (**a**) | (**b**) |

**Figure 13.** 1990 homicide rate eigenvector spatial filters (ESFs). Left (**a**): The PSA ESF ($ESF_{PSA}$). Right (**b**): The NSA ESF ($ESF_{NSA}$). For the ESF measurement scales, relatively speaking, green denotes $y_L$, yellow denotes $y_M$, and red denotes $y_H$.

Table 3 summarizes selected estimation results. Its contents indicated that the uncovered overdispersion was a function of spatial autocorrelation. Three important outcomes here were: (1) The MESF model noticeably outperformed the auto-normal model; (2) a PSA–NSA mixture existed, with noticeable consequences attributable to ignoring NSA; and, (3) the PSA—NSA mixture allowed the retention of a simpler Poisson model specification, eliminating a need to replace it with an assumption of a gamma distributed mean and thus a NB model specification. This last finding was of interest because of concerns raised by Diggle and Milne [32] pertaining to incompatibilities between the presence of non-zero spatial autocorrelation and substitution of the NB for the Poisson probability model.

In conclusion, MESF furnishes a superior model specification in comparison to spatial autoregressive options, in part because it enables a Poisson/NB GLM regression specification that avoids specification error arising from the use of a normal approximation for count data. Evidence exists that the georeferenced crime data analyzed here contain a PSA–NSA mixture, one masked by the dominance of the PSA component. Disregarding the NSA component also leads to serious specification error, namely the use of a NB probability model when a Poisson one is suitable (i.e., a simplicity issue). In other words, acknowledging the presence of NSA matters!

**Table 3.** Parameter estimates for the 1990 US South homicide rates.

| Covariate | Model Specification | | | | |
|---|---|---|---|---|---|
| | Normal Approximation | Poisson | NB | NB + ESF$_{PSA}$ | Poisson + ESF$_{PSA}$ & ESF$_{NSA}$ |
| Resource deprivation/affluence | 0.4239 * | 0.5250 * | 0.4649 * | 0.4733 * | 0.4937 * |
| Population size/density | 0.0806 * | 0.3003 * | 0.2108 * | 0.1825 * | 0.2095 * |
| Median age | −0.0035 * | −0.0104 * | −0.0020 * | −0.0005 * | −0.0015 * |
| Divorce rate | 0.0453 * | 0.0632 * | 0.0588 * | 0.0840 * | 0.0765 * |
| Unemployment rate | −0.0591 * | −0.0731 * | −0.0536 * | −0.0372 * | −0.0463 * |
| Deviance statistic | | 3.1343 | 1.0926 | 1.1897 | 1.0874 |
| Over-dispersion parameter | | | 0.1064 | 0.0256 | 0.0000 |
| (Pseudo)-R$^2$ | 0.281 | 0.311 | 0.308 | 0.476 | 0.596 |

NOTE: * Denotes statistical significance at the 0.01 level; no covariate coefficients are significant at only the 0.05 or 0.10 levels.

## 4. Conclusions; Lessons Learned, and Implications

NSA is an elusive confounder lurking in the background of many spatial statistical analyses. One consequence of ignoring this situation is specification error, as illustrated by the preceding 1990 US South homicide rates (e.g., employing a NB rather than Poisson probability model). NSA conceptualizations are a necessary and superior way of thinking about, especially, contexts involving spatial competition and/or geographic displacement. The synchronization characterizing PSA fails to characterize these spatial correlation contexts. However, NSA is not a phenomenon of spatial data that analysts believe they often encounter. This paper argues that NSA becomes invisible because of a past focus on expanding rather than contracting multilocational firms (e.g., market cannibalization has not been a prominent research theme), the set of common research variable definitions (e.g., nighttime rather than a ratio of daytime-to-nighttime population), and the presence of PSA–NSA mixtures, most seeming to have dominant PSA. These latter two subjects interface in the circumstances of shrinking cities (e.g., Detroit), whose numbers are growing.

Discussions in this paper coupled with the existing paucity of literature addressing NSA imply the need to pursue a research agenda that includes the following issues:

1. Developing appropriate quantification modifications that transform NSA index scales to the interval [−1, 0);
2. Evaluating the impact of different definitions of spatial weights (e.g., topological adjacency, distance, and nearest neighbors), as well as distance standardization [33], on a resulting NSA value;
3. Devising general map pattern descriptions for different degrees of NSA (paralleling the global, regional, and local descriptors for PSA);
4. Revisiting various data analytic features that entail a change of studied variables (e.g., denominators of rates and populations at risk);
5. Articulating relationships between NSA and both geographic scale and resolution, as suggested by the geostatistical wave-hole semi-variogram model and this paper's aggregation experimental results for Detroit;
6. Seeking an informed answer to the question asking whether or not areal unit polygons should be designed to mask or accentuate NSA;
7. Addressing repeatability and replicability of findings by investigating case studies beyond Detroit, the DFW MSA, and the US South with exploratory spatial statistical analysis of other geographic landscapes to see if they, too, exhibit NSA;
8. Expanding findings about the full range of geographic flows beyond the DFW MSA journey-to-work analysis presented here;

9.  Relating MC values in the cross-validation type close-one-store scenario to individual outlet attributes;

10. Replacing Thiessen polygons with Huff probabilities in market area competition analyses;

11. Establishing the range of PSA–NSA mixtures, and further explicating the notion of hidden NSA;

12. Assessing the range of geographic variance accounted for by NSA, specifically to ascertain whether or not 10% is common, and 25% is exceptional;

13. Comprehensively evaluating the strategy of separately estimating $ESF_{PSA}$ and $ESF_{NSA}$ components;

14. Confirming more cases where ignoring NSA results in specification error;

15. Determining the phantom/search degrees of freedom for a given nature and degree of spatial autocorrelation; and,

16. Formulating a better understanding of effective geographic sample size as it relates to phantom/search degrees of freedom.

Most certainly other items could be included in this enumeration. Nevertheless, the discussion in this paper supports these as being among the important future research topics whose pursuits would change the status of NSA from that of being one of the most neglected concepts in spatial statistics

**Funding:** This research received no external funding.

**Acknowledgments:** Daniel A. Griffith is an Ashbel Smith Professor of Geospatial Information Sciences.

**Conflicts of Interest:** The author declares no conflict of interest.

## Appendix A. About PSA-NSA Mixtures and MESF Eigenvector Selection

Stepwise regression variable selection involves phantom and/or search degrees of freedom [34,35], say $p_s$, attributable to the combinatorial selection process that considers variables whether or not they are selected (e.g., considerable multiple testing occurs). Spatial autocorrelation represents redundant information in a variable that effectively reduces n to some smaller number of equivalent independent and identically distributed (IID) values [36]; this effective geographic sample size is the corresponding n for IID values void of spatial autocorrelation. Chun et al. [37] exploit this georeferenced data analytic property to establish a PSA cases formula that determines how many eigenvectors should be included in the candidate set from which selection is made by stepwise regression. As the degree of PSA increases, this candidate set increases (i.e., the effective geographic sample size decreases toward its lower limit of one). The quantities $p_s$ and spatial autocorrelation based effective geographic sample size overlap to some degree: As spatial autocorrelation increasingly is accounted for by eigenvectors, its variance inflation in Y decreases as the accompanying effective geographic sample size moves closer and closer to n, with this increase partly accounting for phantom/search degrees of freedom, causing the resulting z/t-scores to increase, changing their zero spatial autocorrelation null hypothesis, $H_0$, probabilities. For a pure spatial autocorrelation linear regression model specification, the parameter estimates' variance-covariance matrix becomes

$$(\mathbf{X}^T\mathbf{X})^{-1}MSE = \begin{pmatrix} 1/n & \mathbf{0} \\ \mathbf{0}^T & \mathbf{I} \end{pmatrix} \frac{ESS}{n-1-k},$$

where T denotes the matrix transpose operator, $\mathbf{0}$ is a 1-by-k vector of zeroes, $\mathbf{I}$ is the k-by-k identity matrix, MSE denotes mean squared error, and ESS denotes error sum of squares. ESS contains the variance inflation term $TR(\mathbf{V}^{-1})/n$, where TR denotes the matrix trace operator, and $\mathbf{V}$ is a spatial covariance matrix [e.g., $(\mathbf{I} - \varrho\mathbf{W})^T(\mathbf{I} - \varrho\mathbf{W})$ for the simultaneous autoregressive (SAR) model; see §3.4]. This variance inflation factor accompanies the adjustment factor $TR(\mathbf{V}^{-1})/\mathbf{1}^T\mathbf{V}^{-1}\mathbf{1}$ that multiplies n in the matrix to convert it to its effective geographic sample size [36]. As k increases (i.e., a stepwise procedure includes an increasing number of eigenvectors), $\varrho \to 0$, $TR(\mathbf{V}^{-1})/n \to 1$, and $[TR(\mathbf{V}^{-1})/(\mathbf{1}^T\mathbf{V}^{-1}\mathbf{1})]n \to$

n; the MSE decreases (because of k in its denominator, as well as variance accounted for by selected eigenvectors).

The auto-normal maximum likelihood estimate for $\hat{\sigma}^2$ is

$$\mathbf{Y}^{\mathrm{T}}(\mathbf{I} - \mathbf{11}^{\mathrm{T}}/\mathrm{n})(\mathbf{I} - \hat{\rho}\mathbf{W})^{\mathrm{T}}(\mathbf{I} - \hat{\rho}\mathbf{W})(\mathbf{I} - \mathbf{11}^{\mathrm{T}}/\mathrm{n})\mathbf{Y}/\mathrm{n},$$

which is asymptotically unbiased and may be modified to essentially be unbiased by changing its denominator from n to (n − 2). Equating this variance estimate to the revised MESF MSE term

$$\mathrm{ESS}/(\mathrm{n} - 1 - \mathrm{k} - \mathrm{p_s})$$

posits an equation that allows a solution for $p_s$. Table A1 reports integer $p_s$ values for the Detroit and DFW MSA normal random variable case studies appearing in this paper. LASSO estimates [35] essentially corroborate these calculations (the LASSO residuals still contain significant spatial autocorrelation; the DFW MSA LASSO results involve only an NSA ESF). Table A2 entries indicate that adjusting this typically underestimated variance [38] fails to change the $H_0$ probabilities enough to alter the sets of selected eigenvectors when a one-tail $\alpha = 0.10$ test replaces a two-tailed test. Justification for this one-tail replacement of a two-tailed test is that eigenvectors are unique except for a multiplicative factor of −1; in other words, the sign of an eigenvector regression coefficient is arbitrary, converting it to a one-tail test situation. In addition, this is the expected outcome because the residual MC values signify the presence of only trace amounts of spatial autocorrelation. Therefore, the division of $\alpha$ by 2 operates like a Bonferroni type of adjustment. For the Detroit case study, for example, this adjustment retains five PSA eigenvectors that account for only 2.80% of variance, and nine NSA eigenvectors that account for only 4.72% of variance, preventing an increase in the residual $z_{MC}$ from 0.56 to 2.89, which becomes significant (with an accompanying increase in effective geographic sample size). Because the goal of MESF is to account for spatial autocorrelation, elimination of it in residuals should be the prevailing criterion. The difference between the Detroit $R^2$ and predicted-$R^2$ of 0.502–0.320 is bothersome; however, the associated PRESS (predicted residual sum of squares) statistic falls within the 95% confidence interval of the original MESF PRESS statistic.

This foregoing discussion implies that MESF should adopt a liberal implementation significance level for eigenvector selection, such as $\alpha = 0.10$ with a two-tailed test, for empirical analyses involving spatially autocorrelated data. In contrast, IID data should adopt a Bonferroni type of adjustment, where 0.10 is divided by the number of eigenvectors in a candidate set. A very liberal version of this significance level would be 0.01 for only either PSA or NSA eigenvectors and 0.005 when the analysis includes both. Another suggestion by Legendre and his collaborators ([39], p. 359) for addressing this issue is executing two separate stepwise selections, one for PSA eigenvectors, and the other for NSA eigenvectors, and then pooling the two sets of selected vectors for a single final regression; although it should reduce $p_s$ in each estimation situation, being exceptionally ad hoc is a serious weakness of this approach. The topmost question of interest here may be phrased as follows: Especially given that NSA characterizes regression residuals because population parameters are unknown and estimated with sample statistics, is an estimated PSA–NSA mixture ESF portraying heterogeneous spatial autocorrelation (i.e., some individual MC cross-product terms being positive and some being negative, with magnitudes beyond the token ones materializing with genuinely zero spatial autocorrelation) at a given geographic resolution an artifact of the methodology? In other words, does an NSA component emerge strictly to counterbalance part or all of an estimated PSA component, and vice versa? This appendix furnishes evidence to answer this question from a set of simulation experiments, each based upon one of the DFW MSA (aggregated census tracts), Detroit, and US South geographic landscapes studied in this paper in terms of PSA–NSA mixtures. Thus, these experiments have n values of 308, 472, and 1412. In each case, the candidate eigenvector set contains >> 0.5n vectors, implying a need for these simulation experiments [38].

**Table A1.** Summary quantities for determining phantom/search degrees of freedom for constructed ESFs.

| Geographic Landscape | Detroit | DFW MSA |
|---|---|---|
| Standardized response variable variance | 1 | 1 |
| SAR residual variance | 0.69799 | 0.86834 |
| ESF residual variance | 129.69892/(308 − 1 − 48) | 276.43766/(472 − 1 − 51) |
| Estimated search degrees of freedom | 73 | 102 |
| Corrected ESF residual variance | 0.69731 | 0.86930 |
| LASSO based ESF residual variance | 0.68271 | 0.90344 |

**Table A2.** Frequencies of probabilities under $H_0$ of eigenvectors selected in the empirical analyses.

| $H_0$ Probability | <0.0001 | 0.0001–0.0050 | 0.0050–0.0100 | 0.0100–0.0500 | 0.0500–0.1000 |
|---|---|---|---|---|---|
| | | | *PSA* | | |
| Detroit | 4 (2;1) | 6 (7;6) | 0 (1;2) | 8 (8;3) | 5 (5;6) |
| DFW MSA | 0 (0;0) | 0 (0;0) | 0 (0;0) | 10 (9;2) | 6 (7;7) |
| US South | 24 | 41 | 10 | 27 | 8 |
| | | | *NSA* | | |
| Detroit | 2 (0;0) | 2 (2;2) | 1 (2;1) | 11 (12;7) | 9 (9;6) |
| DFW MSA | 1 (1;1) | 11 (4;1) | 6 (6;3) | 14 (20;17) | 3 (4;9) |
| US South | 7 | 43 | 9 | 38 | 5 |

NOTE: Frequencies in parentheses (two-tailed; one-tail) tabulate those eigenvector selections based upon a corrected variance estimate with a denominator that includes search degrees of freedom.

The design of the simulation experiments entailed 10,000 replications, shrinking the PSA range to (0, 1] and stretching the NSA range to [−1, 0) to select eigenvectors for the candidate sets, normal random variables with $\mu = 0$ and $\sigma^2 = 1$ for which the probability of the Shapiro-Wilk [i.e., P(S-W)] diagnostic statistic implied a failure to reject the null hypothesis (to avoid contamination by one obvious source of specification error), and linear regression variable selection significance levels of 0.005 and 0.010. These significance levels represented the extremely liberal Bonferroni adjusted probability thresholds that compensated for the phantom/search degrees of freedom, in the presence of zero spatial autocorrelation, of $0.10/308 \approx 0.00032$, $0.10/472 \approx 0.00021$, and $0.10/1471 \approx 0.00007$. The candidate eigenvector sets were determined by $MC_j/MC_{max} > 0.25$ for PSA vectors, and $MC_j/|MC_{min}| < −0.25$ for NSA vectors. The number of NSA eigenvectors in a candidate set was, respectively, about 70%, 80%, and 90% more than the number of PSA eigenvectors. Table A3 summarizes output from these simulation experiments.

Table A3 reports summary statistics for the pseudo-random numbers implying that the simulated normal random variables were good. Implications of simulation findings tabulated in this table included the following: (1) The expected and average number of selected false-positive eigenvectors were consistent; (2) $\overline{R^2}$ for p = 0.005 was negligible but was somewhat large for p = 0.01, given the 5% rule; and, (3) no evidence of overfitting (PRESS/ESS $\approx 1$). These results bolster the PSA–NSA mixture verdicts presented in this paper, especially given the number of selected eigenvectors with extremely small probabilities (see Table A2) appearing in the three empirical examples.

Table A1 suggests that the number of phantom/search degrees of freedom was sizeable, and perhaps about 34% of the candidate set of eigenvectors in the linear regression case studies. This finding was consistent with claims appearing in Freedman et al. [38].

Table A2 further suggests that many selected eigenvector $H_0$ probabilities were extreme enough that increasing an MSE by subtracting more degrees of freedom in its denominator failed to make them nonsignificant when changing from a two-tailed to a one-tail test using the same $\alpha$. This table also suggests that mixtures may well tend to have a number of both PSA and NSA eigenvectors.

**Table A3.** Summary statistics from the simulation experiments (ranges are in parentheses); a normal random variable with μ = 0 and σ² = 1, stepwise regression eigenvector selection, and 10,000 replications.

| Statistic | $\hat{\mu}$ | $\hat{\sigma}$ | $\overline{P(S\text{-}W)}$ | $\overline{\text{# vectors}}$ | Candidate set Size | $\overline{R^2}$ | $\overline{\text{PRESS/ESS}}$ |
|---|---|---|---|---|---|---|---|
| | | | | $\alpha = 0.005$ | | | |
| Detroit | −0.000 (−0.223, 0.207) | 0.999 (0.833, 1.156) | 0.548 (0.1001, 0.9998) | 1.228 (0, 10) | 80 + 135 = 215 | 0.037 (0, 0.263) | 1.015 (1.007, 1.075) |
| DFW | 0.001 (−0.171, 0.175) | 1.000 (0.872, 1.121) | 0.545 (0.1000, 0.9999) | 1.701 (0, 10) | 105 + 193 = 298 | 0.034 (0, 0.180) | 1.012 (1.004, 1.043) |
| US South | 0.000 (−0.097, 0.109) | 0.999 (0.932, 1.071) | 0.543 (0.1002, 0.9995) | 5.903 (0, 17) | 352 + 662 = 1,014 | 0.039 (0.017, 0.107) | 1.010 (1.001, 1.035) |
| | | | | $\alpha = 0.010$ | | | |
| Detroit | 0.000 (−0.219, 0.197) | 0.999 (0.844, 1.142 | 0.546 (0.1002, 0.9999) | 2.690 (0, 13) | 80 + 135 = 215 | 0.070 (0, 0.292) | 1.025 (1.007, 1.100) |
| DFW | −0.000 (−0.183, 0.176) | 0.999 (0.891, 1.113) | 0.547 (0.1001, 0.9997) | 3.618 (0, 15) | 105 + 193 = 298 | 0.062 (0, 0.242) | 1.020 (1.004, 1.074) |
| US South | −0.000 (−0.100, 0.091) | 1.000 (0.932, 1.068) | 0.542 (0.1001, 0.9999) | 12.870 (1, 30) | 352 + 662 = 1,014 | 0.073 (0.005, 0.164) | 1.020 (1.003, 1.047) |

NOTE: P(S-W) denotes the probability of the Shapiro-Wilk normality diagnostic statistic. PRESS denotes prediction error sum of squares (ESS).

## Appendix B. The Geographic Distribution of the Spatial Means of the Census Tract Centroids

The population weighted coordinates revealed a close correspondence between store locations and spatial means.

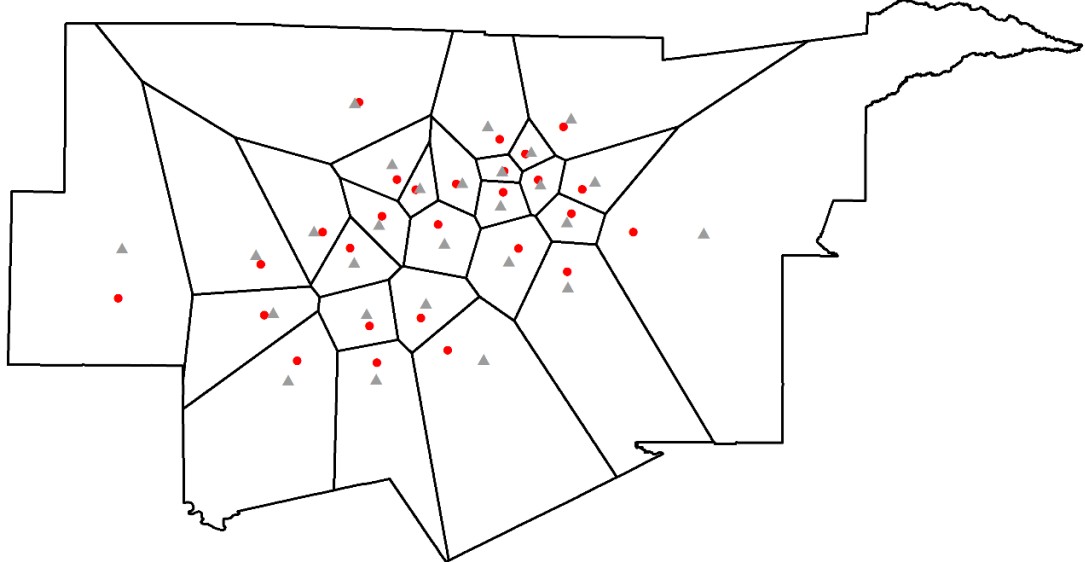

**Figure A1.** The Thiessen polygon delimited market areas of all 27 Kohl's stores. Solid red circles denote the store locations. Solid gray triangles denote the spatial means of the census tract centroids, weighted by population.

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
