# Peer review of "Negative Spatial Autocorrelation: One of the Most Neglected Concepts in Spatial Statistics"

_stats, doi:10.3390/stats2030027_

Round 1

Reviewer 1 Report

This article is of undeniable interest, and is very interesting. Nevertheless, I have a few remarks to make:

- Overall, the article seems to me to be too long and rather uneven, with very common statistical concepts being followed by very specific considerations. It follows a plan that is quite confusing for the reader, and the reader does not really see where the author is going with it: it is necessary to wait for paragraph 1.3 and especially the conclusion to have the list of questions that would deserve further development and correspond to the title of the article. None of these questions are really addressed in the article: after a very long introduction and detailed examples, the reader remains a little hungry.

- Beyond the questions formulated in the conclusion, some other questions remain poorly detailed, in particular the influence of the definition of spatial weights (spatial weights make a considerable difference with "classical" correlations, which makes reminders of classical statistical correlations somewhat irrelevant), scale effects, masking effects, and more generally everything that can be reproached for spatial autocorrelation indices.

- differences in approach in spatial weights are not really discussed (neighbourhood, distances, adjacency, etc.), as well as their influence on PSA and NSA,

- the influence of the stationarity hypothesis is not really discussed, although it remains at the heart of the use of global indices.

- as indicated in the article, the NSA is more difficult to obtain than the PSA, for obvious reasons in the continuing case. As whit PSA, it would be interesting to try to characterize the spatial pattern of the NSA, beyond the assertion that it corresponds to a competitive effect as shown in the examples.

- how to separate, in the same index, the PSA and NSA components? Could new indices be defined on this basis?

- the examples presented are interesting, but seem too detailed to me. Knowing the unresolved questions about the NSA (as stated in the conclusion) and showing what their answers would bring to the spatial analysis would be very educational.

Some other detailed remarks:

- the main autocorrelation indicators (Moran, Geary) can be expressed in , 4, a simpler way (cf Souris M, DEmoraes F. Improvment of Spatial Autocorrelation, Kernel estimation, and Modeling Methods by Spatial Standardization on Distance. Int.. J. Geo-Inf., 2019, 8(4), 199)

- spatial weights do not intervene in the average of the indices (under null hypothesis), but they do intervene in their variance: how to separate the variance due to the NSA from the variance due to the PSA?

Author Response

This reviewer states that the paper is “too long,” but askes for considerable additions to it. The paper can be shortened a bit by removing Appendix A (and references to it: l 392, 482); I have not done this, and leave it as an editorial decision.

An insertion in the future research agenda (point #2: l 833-835) presents the need to evaluate impacts of different definitions of spatial weights. Point #5 already notes a need to assess geographic scale and resolution.

Five mentions of stationarity (l 384, 522, 528, 660, 734) already exist. Expanding upon this topic would increase the length of the paper.

An insertion in the future research agenda (point #3: l 836-837) presents the need to devise general map pattern descriptors for generic degrees of NSA.

The unresolved questions furnish at least a partial future research agenda.

Proposing new indices, and decomposing the variance contributions of PSA and NSA in indices are beyond the scope of this paper; they could constitute a lengthy paper on their own.

A citation now appears to Souris and Demoraes (2019) on l 834.

Reviewer 2 Report

A brief summary:

This manuscript focuses on the concept of negative spatial autocorrelation
(NSA), its properties and the reasons why it is often neglected in spatial statistics such
as quantification difficulties, variables studied and mixture of positive and negative autocorrelation.
Four examples are then developed and presented to illustrate the concept.
These include market area competition, shifts in daytime and nighttime
populations, urban area shrinkage and homicide rates.
Finally, author concludes the paper by suggesting the issues that need to be

Broad comments:

The question of interest is well defined, illustrated and clearly of great
interest to the readers. However, in its present form parts of the manuscript
seem to be not entirely innovative. The author is an expert in the field and has a large number of publications
explaining why statisticians should care about NSA, for example [1] and [2].
However, the references to those publications are
not provided, and it is not entirely clear how present manuscript
is different/similar to the previous ones.
Having said that, the results do provide an
advance in current knowledge, which is the strength of the manuscript.

Quality of presentation is overall very good; it is a well written manuscript.
In some places (for example section 1.2 and 1.3), however, text can be improved by introducing lists of items,
which will help readers to better follow the line of reasoning.

Specific comments:

L37 'tend to pair with median values of Y'
Should it not be 'medium' similar to the rest of the text?
L130 'Texas percentage urban population and population density'
Should it not be 'rural'?
L448-471 Steps need to be clearly written and related to each other.  
L577 'workers travelling to census tracts to their jobs'
Should it not be 'from their jobs'?

References:

[1] Griffith, D. A., & Arbia, G. (2010). Detecting negative spatial autocorrelation
in georeferenced random variables. International Journal of Geographical
Information Science, 24(3), 417–437. doi:10.1080/13658810902832591

[2] Griffith, D. A. (2006). Hidden negative spatial autocorrelation. Journal of
Geographical Systems, 8(4), 335–355. doi:10.1007/s10109-006-0034-9

Author Response

The two Griffith citations are included now (they already were included indirectly with the citation of [12]), accompanied by a description differentiating this paper from them (l 73-76).

§1.2 and 1.3 now begin with lists of items (l 145-151 & l 249-255).

The first two noted mistakes are corrected; the fourth item (“… to their jobs”) is not a mistake. Clarifications have been added (l 512-519) to address the third item.